# Involvement of DAAO Overexpression in Delayed Hippocampal Neuronal Death

**DOI:** 10.3390/cells11223689

**Published:** 2022-11-21

**Authors:** Hao Liu, Jun-Tao Zhang, Chen-Ye Mou, Yue Hao, Wei Cui

**Affiliations:** 1Department of Ophthalmology, The Affiliated People’s Hospital of Ningbo University, Ningbo 315040, China; 2School of Medicine, Ningbo University, Ningbo 315211, China

**Keywords:** D-amino acid oxidase, middle cerebral artery occlusion, neuronal death, transient receptor potential channel, hydrogen peroxide

## Abstract

Background: D-amino acid oxidase (DAAO) is a flavoenzyme that specifically catalyzes the deamination of many neutral and basic D-amino acids. This study aims to explore the pathological increment of hippocampal DAAO and its potential relationship with delayed hippocampal neuronal death. Methods: Ischemia–reperfusion was induced in mice through middle cerebral artery occlusion (MCAO). Neurological deficit scores and hippocampal neuronal death were assessed in MCAO mice. Immunofluorescent staining was applied to identify activated astrocytes and evaluate DAAO expression. TUNEL and Nissl staining were utilized to identify cell apoptosis of hippocampal neurons. Results: Hippocampal astrocytic DAAO was strikingly increased following ischemic stroke, with the greatest increase on day 5 after surgery, followed by the manifestation of neurobehavioral deficits. Astrocytic DAAO was found to be mainly expressed in the hippocampal CA2 region and linked with subsequent specific neural apoptosis. Thus, it is supposed that the activation of astrocytic DAAO in ischemic stroke might contribute to neuronal death. An intravenous, twice-daily administration of 4*H*-furo[3,2-b]pyrrole-5-carboxylic acid (SUN, 10 mg/kg) markedly relieved behavioral status and delayed hippocampal neuronal death by 38.0% and 41.5%, respectively, compared to the model group treated with saline. In transfected primary astrocytes, DAAO overexpression inhibits cell activity, induces cytotoxicity, and promotes hippocampal neuronal death at least partly by enhancing H_2_O_2_ levels with subsequent activation of TRP calcium channels in neurons. Conclusions: Our findings suggest that increased hippocampal DAAO is causally associated with the development of delayed neuronal death after MCAO onset via astrocyte–neuron interactions. Hence, targeting DAAO is a promising therapeutic strategy for the management of neurological disorders.

## 1. Introduction

D-Amino acids are important neurotransmitters that play key functions in the nervous system. The flavoprotein D-amino acid oxidase (DAAO) is the main metabolic enzyme of neutral and basic D-amino acids. The physiological roles identified for DAAO include regulation of the nervous system and hormone secretion, among others, as well as involvement in the development of different diseases resulting from alterations in DAAO activity. In the central nervous system, the pathophysiological functions of DAAO are implicated in hyperalgesia, morphine antinociceptive tolerance, and cognitive impairment, such as in the context of schizophrenia or other neurodegenerative diseases [1,2,3,4]. In postmortem autopsies, hippocampal DAAO levels are significantly higher in schizophrenia patients than in people without schizophrenia and are closely linked with disease course. Moreover, the expression of DAAO is more than 70% higher in long-term (>20 years) schizophrenia patients than people without schizophrenia [5]. Hence, DAAO inhibitors have been applied in schizophrenia therapy in a number of subsequent clinical studies [6,7]. In 2013, the first clinical report of DAAO inhibitor sodium benzoate use in the treatment of patients with chronic schizophrenia was published. Six weeks of treatment significantly improved patient cognition and alleviated other symptoms, and the drug was well tolerated (clinical trial registration number: NCT 00960219). A randomized, double-blinded, placebo-controlled clinical trial found that peripheral DAAO levels in dementia patients with psychobehavioral symptoms (BPSDs) are positively correlated with Alzheimer’s Behavioral Scale scores. DAAO inhibitors could significantly improve cognitive function in patients with BPSDs (clinical trial registration number: NCT02103673) [8]. It has been demonstrated that DAAO expression is upregulated in a variety of neurodegenerative diseases, and DAAO-targeted therapies have resulted in improvements regarding clinical symptoms.

Our previous report was the first to disclose that DAAO expression steadily increases in activated astrocytes after ischemic stroke onset [9]. DAAO inhibitors or siRNA/DAAO could effectively alleviate brain injury by suppressing levels of the byproduct hydrogen peroxide, which implies that DAAO might be involved in the process of ischemic stroke. D-Ser is found at high physiological levels in the cerebrum and can be degraded by DAAO to increase levels of the byproduct hydrogen peroxide, which is one of the most stable reactive oxygen species (ROS), thus causing cytotoxicity to induce neuronal apoptosis as well as to reduce the expression of brain-derived neurotrophic factor (BDNF) [10,11]. In our previous study, we reported that cortical DAAO is upregulated following MCAO and is predominantly expressed in astrocytes. However, the cell specificity of hippocampal DAAO expression remains unknown. In addition, application of the DAAO inhibitor SUN reduces neurological deficits following MCAO, but the specific downstream mechanisms for this remain unclear. In this study, we focus on astrocytic DAAO expression in the hippocampus and whether it is related to delayed neuronal death after ischemic insult, which also needs clarification regarding astrocyte–neuron interactions. Furthermore, we subsequently investigate whether astrocytic DAAO in the hippocampus influences neurological deficits through its effect on neuronal survival. This study uses fluorescent probes to explore neurotoxicity and its underlying mechanism of hippocampal DAAO induction in response to acute neuropathological insult. Here, pcDNA3.1(+)-*daao*-transfected primary astrocytes are used to mimic the pathological overexpression of astrocytic DAAO in order to explore the causal relationship between DAAO and delayed neuronal death.

## 2. Methods

### 2.1. Chemicals

The compound SUN (4H-furo[3,2-b]pyrrole-5-carboxylic acid) was synthetized by ENNO Bioscience (Shanghai, China). SKF96365, N-tert-butyl-α-phenylnitrone (PBN), and D-kynurenine were purchased from Sigma-Aldrich (St. Louis, MO, USA). Flavine adenine dinucleotide (FAD), kynurenic acid (KYNA), and bovine serum albumin were obtained from Aladdin (Shanghai, China). Trichloroacetic acid and dimethylbenzene were purchased from the Sinopharm group (Beijing, China). All drugs and reagents were dissolved in saline.

### 2.2. Animals

Specific-pathogen-free Swiss mice (male, >8 weeks, 20–22 g) were purchased from the Shanghai Experimental Animal Institute for Biological Sciences (Shanghai, China). The mice were reared in cages with free access to food and water following the animal care guidelines of the NIH (Bethesda, MD, USA). The animal houses used have an authorized biosafety license and were programmed to ensure a temperature-controlled environment with a normal 12 h light–dark cycle. The animal protocols were approved by the Animal Care and Welfare Committee of Shanghai Jiao Tong University (Shanghai, China) and were performed in accordance with the Guide for Care and Use of Laboratory Animals and the Animals in Research: Reporting In Vivo Experiments guidelines [12,13].

### 2.3. Mouse MCAO Model

As the most popular ischemic stroke model, transient MCAO was executed using a modified intraluminal occlusion technique with a monofilament as in our previous report [9]. In brief, mice were intraperitoneally injected with 1.5% pentobarbital sodium (Aladdin Biochemical Technology, Shanghai, China) before surgery. After the common carotid artery was exposed and ligated to stop bleeding, a 6-0 nylon monofilament coated with a silicone tip (Rayward Life Technology, Shenzhen, China) was inserted into the common carotid artery to a position about 0.8–0.9 cm deep starting from the bifurcation site for occlusion of the middle cerebral artery. The monofilament was fixed to block the middle cerebral artery for 90 min in an incubator at 32 ± 0.5 °C and withdrawn for subsequent reperfusion. The exclusion criteria were as follows: (1) unexpected death; (2) no infarct injury or only a small infarct size; and (3) intracranial hemorrhage [14]. Following implementation of these criteria, around 60% to 80% mice remained. All ischemic mice were randomly divided into each group (*n* = 4–6), as indicated in the legend figures, according to simple randomization based on body weight (range from 20 to 22 g). The mice in the sham group were subjected to the same surgical process without inserting the intraluminal monofilament for 5 days following pseudo-surgery. Approximately 20% of the ischemia–reperfusion-induced mice were excluded due to death on days 4 and 5 after surgery. The following steps were undertaken to study the involvement of hippocampal DAAO in ischemic strokes: (1) Many indicators were measured to assess the relationships between DAAO expression and pathological changes in MCAO mice on days 0, 1, 3, and 5 after ischemia–reperfusion. These subgroups were defined according to the timepoint after surgery. Normal mice were defined as MCAO mice on day 0. (2) The DAAO inhibitor SUN was administrated to further evaluate efficacy. The mice were divided into a sham group, an MCAO control group (MCAO mice treated with saline only), and an MCAO treatment group (MCAO mice administered 10 mg/kg DAAO inhibitor SUN in intravenous, twice-daily injections). All of the described assays were conducted on samples from this study.

### 2.4. Behavioral Deficit Score

Neurobehavioral status was evaluated after ischemic stroke using the Zea Longa neurological grading scale (0–5) [15]. Grades were scored by the deficit degree of hemiparesis in contralateral limbs: 0 = no observable deficit; 1 = flexural forepaw on ipsilateral side; 2 = walking in circle; 3 = falling when walking; and 4 = unable to walk with depression status.

### 2.5. Immunofluorescence Staining in Brains

To assess cell-specific expression of hippocampal DAAO, cerebral sections (30 μm thickness) were collected from MCAO mice on day 5 after surgery and prepared as frozen sections. As described in our former report [9], mice were intracardially perfused with 20 mL saline followed by 20 mL 4% (*w*/*v*) paraformaldehyde solution before isolation of the cerebrum. Then, brains were fixed in 4% buffered paraformaldehyde solution for the whole night and dehydrated in gradient sucrose solutions (10–30%) at 4 °C. Cerebral tissue was entrapped by freezing in an optimal-cutting-temperature embedding agent (Leica Microsystems) before cutting into sections (30 μm thickness). The hippocampal sections were selected and incubated in 10% goat serum (*v*/*v*) including 0.5% Triton X-100 (*v*/*v*) in PBS for 1 h before incubation with antibodies. Cerebral sections were dried in an oven at 37 °C for 40 min to retrieve tissue antigens. After washing with 0.05 M PBS three times and incubating in sealing fluid (10% goat serum (*v*/*v*) containing 0.5% Triton X-100 (*v*/*v*)) for 1 h, sections were subsequently incubated with DAAO antibody (1:100, tailored polyclonal antibody from Novus Biologicals) and other primary antibodies as different cytological markers (anti-Iba-1 for microglia: 1:50, mouse monoclonal, Millipore; anti-GFAP for astrocytes: 1:100, mouse monoclonal, Millipore; anti-nestin for reactive astrocytes: 1:200, mouse monoclonal, Proteintech; anti-NeuN for neuron: 1:50, mouse monoclonal, Millipore) for an additional 15–18 h at 4 °C. After washing with 0.05 M PBS, the immunocolocalizations of DAAO and each cytological marker were fluorescence-labeled with Alexa-555-conjugated (1:200, goat anti-rabbit, Invitrogen, Carlsbad, CA, USA) and Alexa-488-conjugated (1:200, goat anti-mouse, Invitrogen, Carlsbad, CA, USA) secondary antibodies. Immunological colocalization of hippocampal DAAO with microglia, astrocytes, or neurons was visually assessed under a TCS SP8 confocal microscope (Leica Microsystems, Wetzlar, Germany) [9].

To examine time courses of immunolabeled DAAO in the hippocampal CA2 area, cerebral slices were collected from MCAO mice on days 0, 1, 3, and 5 after ischemia–reperfusion. To quantify the immunofluorescent intensity of cerebral slices in the hippocampal CA2 area, fields of merged images were randomly selected under a confocal microscope with 20× or 40× magnification, and ImageJ software (National Institutes of Health, Bethesda, MD, USA) was applied blindly by an investigator to measure the colocalized pixels of immunostaining areas accordingly. The quantified area fractions are represented as the ratio of the colocalized area to the total scanned area.

### 2.6. Western Blot

Samples in these tests were extracted from the hippocampal tissue of MCAO mice on days 0, 1, 3, and 5 after ischemia–reperfusion or from pcDNA3.1-transfected primary astrocytes. Western blotting was conducted to detect DAAO expression of hippocampal sections or primary astrocytes, as previously described [9]. Samples were extracted using RIPA buffer containing a cocktail of phenylmethylsulfonyl fluoride protease inhibitor and phosphatase inhibitor (50×, Beyotime Institute of Biotechnology, Nantong, China), prepared for electrophoresis by mixing with loading buffer (5×, Beyotime Institute of Biotechnology, Nantong, China), and denatured in a metal bath at 105 °C for 12 min. The protein concentration in the homogenate was determined with a bicinchoninic acid protein method (Beyotime Institute of Biotechnology, Nantong, China). The amount of total protein in the loading sample was 30–50 μg. DAAO antibody (1:3000, tailored polyclonal antibody from Novus Biologicals, Centennial, CO, USA) was used to label the target protein, and β-actin antibody (1:5000, Santa Cruz Biotechnology, Dallas, TX, USA) was applied as a reference protein. After fluorescence-labeling with secondary antibodies (Dylight 680-conjugated anti-mouse IgG (1:10,000) and Dylight 800-conjugated anti-rabbit IgG (1:10,000); Cell Signaling Technology, Danvers, MA, USA), protein bands were visualized using an Odyssey Infrared Imaging system (Li-Cor Biosciences, Lincoln, NE, USA) and quantified with ImageJ software (National Institutes of Health, Bethesda, MD, USA).

### 2.7. DAAO Activity Assay

The hippocampal DAAO activity of MCAO mice was further observed through KYNA and hydrogen peroxide, both of which are byproducts of the oxidative deamination of D-kynurenine catalyzed by DAAO. In accordance with our previous method [9], an aliquot of hippocampal homogenate in 0.4 M Tris-buffer (pH 8.5) was centrifuged at 4 °C to obtain the supernatant as a DAAO source for both the KYNA assay and hydrogen peroxide detection.

KYNA assay: Hippocampal homogenate supernatant (35 μL of 5–8 μg/μL) was incubated with FAD solution (2 mM, 5 μL) for 5 min at 37 °C. Subsequently, D-kynurenine (10 mM, 10 μL) in 0.4 M Tris-buffer (pH 8.5) was added to prepare the mixture for metabolizing D-kynurenine at 37 °C for 3 h. The DAAO enzymatic reaction was terminated by adding 50% trichloroacetic acid (5 μL). After mixing and centrifugation, the supernatant of the reaction solution (50 μL) and zinc sulfate (300 mM, 20 μL in ddH_2_O) were added to 300 μL of 0.4 M Tris-buffer (pH 8.5) for the final fluorescence assay (excitation wavelength of 251 nm and emission wavelength of 396 nm) with an EnSpire 2300 Multimode Plate Reader (Perkinelmer Co., Waltham, MA, USA). The DA (DAAO activity) was calculated as K/M/T (K = A × 370 μL, where A is amount of KYNA according to the standard treatment of the relationship between the optical density and concentration of KYNA; M is the mass of the total protein and is equal to the concentration of total protein × 35 μL; and T is the duration of the reaction, where T = 3 h × 60 = 180 min). The final DAAO activity was indicated in the form of the production rate of KYNA (fmol KYNA/μg protein/min) after calibration of the standard curve.

Hydrogen peroxide detection: Hippocampal homogenate supernatant and astrocyte-conditioned medium were tested using an aqueous hydrogen peroxide assay kit (Sangon Biotech, Shanghai, China). In brief, samples of supernatant diluted by 20 times were incubated with the reaction solution at room temperature for 0.5–1 h. The OD_590nm_ values were recorded using a Varioskan Flash spectral reader (Termo Labsystems, Philadelphia, PA, USA). The hydrogen peroxide levels were normalized and quantified on the basis of a standard curve [1].

### 2.8. TdT-Mediated dUTP Nick-End Labeling (TUNEL) and Nissl Staining

Coronal sections of the hippocampus were prepared as mentioned in Section 2.5. Coronal sections (30 μm thickness) were first prepared by deparaffinizing and rehydrating before staining in toluidine blue solution for 30 min at 60 °C. Then, slides were washed with distilled water and ethanol gradients (70%, 80%, 95%, and 100%) for 1 min in succession. Pure dimethylbenzene was added to enhance light transmission, and coverslips were fixed with neutral balsam. In subsequent applications of an optical microscope, numbers of total neurons (400× field) in the hippocampal CA2 region were counted and calculated blindly by an investigator. TUNEL immunostaining was studied according to the instructions of the TUNEL cell apoptosis detection kit (Beyotime Institute of Biotechnology, Nantong, China). For Nissl staining, the percentage of injured cells = number of injured cells/(number of injured cells + number of normal neurons) × 100.

### 2.9. Isolation of Primary Cells

After fixing brains of one-day-old postnatal pups, the meninge was quickly peeled off using ophthalmic tweezers under a low-power inverted microscope on an ultra-clean platform. One side of the cerebrum was everted gently from the longitudinal fissure near the upper cerebellum. A large number of ependymal attachments were observed near the cerebellum; then, the hippocampus could be recognized and was immediately separated. Mouse hippocampi were minced and digested with 0.05% trypsin (Beyotime Institute of Biotechnology, Nantong, China) for 0.5 h. After screening with 70 and 40 μm filters, dispersive cells were neutralized with 5 mL of complete Dulbecco’s modified Eagle’s medium (Gibco DMEM, Life Science Inc, Woodland Hill, CA, USA) containing 10% fetal bovine serum (FBS, *v*/*v*) and 1% penicillin/streptomycin and centrifuged (600× *g*) for 5 min before resuspension. The cellular suspension was seeded in a 100 μg/mL poly-L-lysine-coated cell culture flask (1 × 10^6–7^) and co-cultured for 10 days. Confluent cultures were shaken at 1× *g* overnight, and culture medium was changed to remove microglia and neurons. DMEM was used to wash culture flasks and remove unattached cells. Oligodendrocytes were detached with 0.05% trypsin at 37 °C for 15 min, and the suspension was eliminated. The remaining purified astrocytes were cultured in complete culture medium and plated for experiments.

For the collection of primary neurons, suspended cells collected by shaking confluent cultures overnight were incubated in complete DMEM for 2 h. Then, neurobasal medium (Invitrogen, Carlsbad, CA, USA) containing Gibco B-27 supplement (50×, Life Science Inc., Woodland Hill, CA, USA) and glutamine (0.5 mM, Life Science Inc, Woodland Hill, CA, USA) was used to perform selective neuronal culture for 5 days. In this neurobasal medium, glial cells were inhibited, and the remaining cells were identified as purified neurons.

### 2.10. Construction of pcDNA3.1(+)-daao and Transfection in Primary Astrocytes

The method in detail was performed as in previous reports [16,17]. *daao* cDNA was obtained from the hippocampal total mRNA of Swiss mice (MagExtractor-RNA kit, Toyobo Co., Osaka, Japan) after reverse transcription using a ReverTra Ace qPCR RT-Kit (Toyobo Co., Osaka, Japan) and was amplified using the following primers: *Hindaao*-F: 5′-TTTAAGCTTatgcgcgtggccgtgatc-3′ and *Ecodaao*-R: 5′-TTTGAATTCtcagaggtgggagggaggcaac-3′. Then, the *daao* coding sequence was introduced into the *Hin*dIII and *Eco*RI multicloning sites of expression vector pcDNA3.1. The transcription of *daao* was driven by CMV promoter. After culturing for 6 days, primary hippocampal astrocytes were transfected with 60 nM *daao* cDNA cloned in pcDNA3.1 or empty vectors (Invitrogen; Thermo Fisher Scientific, Inc., Waltham, MA, USA) using LipoMax transfection reagent (Sudgen, Nanjing, China) and cultured at 37 °C for 2 days. The *daao* mRNA levels and DAAO protein expression were confirmed by real-time quantitative polymerase chain reaction (RT-qPCR) and Western blot, as mentioned above. pcDNA3.1(+)-*daao*-transfected primary astrocytes were utilized to mimic the pathological overexpression of astrocytic DAAO. Astrocyte-conditioned medium (ACM) was transferred into culture plates seeded with primary neurons to explore causal relationships with delayed neuronal death and potential mechanisms.

### 2.11. RT-qPCR

Total mRNA was extracted from pcDNA3.1-transfected primary astrocytes and reverse-transcribed using a ReverTra Ace qPCR RT kit (Toyobo, Osaka, Japan) [18]. RT-qPCR amplification was conducted with the following primers: DAAO: 5′-CTGCATTCATGAGCGTTACC-3′ and 5′-CAGCGTTTGGAGAATGGAGG-3′; GAPDH: 5′-CAAGATTGTCAGCAATGCATCCTG-3′ and 5′-CCTGCTTCACCACCTTCTTGA-3′). cDNA was quantified following fluorescent labeling by incubation in Realmaster Mix (SYBR Green I) (Toyobo, Osaka, Japan). Melting curves were used to confirm the specificity of target cDNA. Quantitative analysis was conducted using the 2^−ΔΔCt^ method after normalizing to GAPDH.

### 2.12. Apoptosis, Cell Viability, and Lactate Dehydrogenase (LDH) Cytotoxicity Assay

After D-Ala (0.3 mM or 1.0 mM) and 2.5 μM FAD were added in culture media with pcDNA3.1-transfected astrocytes and incubated for 6 h, ACM was transferred into culture plates seeded with primary neurons for subsequent studies. For the apoptosis study, primary neurons (1 × 10^5^) were seeded into a 100 μg/mL poly-L-lysine-coated 96-well plate for 4–5 days at 37 °C in a 5% carbon dioxide incubator and incubated with ACM for 2 or 4 days. ACM was renewed every 24 h, and Gibco B-27 supplement (50×, Life Science Inc, Woodland Hill, USA) and glutamine (0.5 mM, Life Science Inc, Woodland Hill, CA, USA) were added to ACM to support neuronal growth. Apoptosis was evaluated using a TUNEL fluorescent probe detection kit (Beyotime Institute of Biotechnology, Nantong, China). Images were obtained with a fluorescence microscope (Nikon Instruments Inc., Melville, NY, USA) at 100× magnification. To quantify the percentage of apoptotic neurons, photos were captured for five random fields of each well, and TUNEL-positive cells were counted and averaged.

In addition, a CCK-8 assay kit (Beyotime Institute of Biotechnology, Nantong, China) was applied to measure the level of yellow formazan dye produced by dehydrogenase in the living cells for cell viability. A portion of neurons (5 × 10^3^ cells/mL) was inoculated with complete DMEM in 96-well plates and maintained in an incubator at 37 °C for one day. Subsequently, the culture medium was replaced with ACM, and neurons were co-cultured with other chemicals (50 μM SUN compound, 10 μM SKF96365, or 10 μM PBN) for one day at 37 °C. Lastly, CCK-8 reagent was incubated with neurons in each well at 37 °C for 2–3 h. OD_450nm_ values were detected with a Multimode Plate Reader (EnSpire 2300, Perkinelmer Co., Waltham, MA, USA).

LDH cytotoxicity was measured using an LDH assay kit following the manufacturer’s instructions (Beyotime Institute of Biotechnology, Nantong, China). After incubation with ACM and other chemicals for one day, the medium was removed and replaced with 150 μL PBS-diluted LDH release reagent. The mixture was shaken gently for 10 s and incubated in a cell incubator for 1 h at 37 °C. After shaking at 500× *g* for 5 min, 150 μL of supernatant was transferred to a new 96-well plate for determination. The test was based on the content of LDH released into the culture medium by damaged neurons. Absorbance at the wavelength of 490 nm was recorded with a Multimode Plate Reader (EnSpire 2300, Perkinelmer Co., Waltham, MA, USA). The results were presented in the form of percentages by calculating the contents of LDH released from neurons in different groups compared to the control group: cell cytotoxicity (%) = (OD_test group_ − OD_control group_)/(OD_test group_ − OD_blank_) × 100%.

### 2.13. Statistical Analysis

Shapiro–Wilk normality testing was conducted to evaluate the distribution of data. Data following non-normal distributions are presented in the form of medians and interquartile ranges and were analyzed via a nonparametric equivalent. One-way ANOVA with a post hoc Student–Newman–Keuls test was applied for time-course-related experiments, TUNEL-labeled apoptotic tests, cell activity assays, and cytotoxicity assays. Unpaired two-tailed Student’s *t*-test was used for the evaluation of DAAO expression in pcDNA3.1(+)-*daao*-transfected astrocytes as well as treatment-related experiments for the prevention of neurobehavioral deficits and delayed hippocampal neuronal death after MCAO surgery. Normally distributed data are presented as means ± SE and analyzed using GraphPad Prism 7 (GraphPad Software, Inc., San Diego, CA, USA). Statistical significance was defined as a *p*-value less than 0.05.

## 3. Results

### 3.1. Cell Specificity of Hippocampal DAAO Expression in Ischemic Mice Induced by Transient MCAO

Considering that both microglia and astrocytes are activated and sustained for many days in ischemic strokes [19,20], the timepoint of day 3 after surgery was used to clarify the cell specificity of hippocampal DAAO in ischemic mice by immunocolocalizing with Iba-1, NeuN, or GFAP, respectively. As shown in the photomicrographs, DAAO-positive cells were extensively labeled with GFAP, whereas no or few DAAO-positive cells were colabeled with Iba-1 or NeuN (Figure 1A–C). The area fractions of DAAO colocalizing with three cellular markers are quantified in Figure 1D. The results suggest that hippocampal DAAO is mainly expressed in activated astrocytes in ischemic stroke associated with complex morphological changes, including cellular hypertrophy. Subsequently, immunocolocalization of double-labeled GFAP/DAAO was also quantified at four timepoints (0 h, 24 h, 3 days, and 5 days) following MCAO surgery using a computer-assisted image analysis program (0.025 ± 0.003%, 0.075 ± 0.013%, 2.76 ± 0.28%, and 3.65 ± 0.20%, respectively) (Figure 1E–P).

### 3.2. Time Course of Hippocampal DAAO Expression and Related Activity after Acute Ischemic Insult

Western blotting was employed first to disclose hippocampal DAAO expression following MCAO injury (relative to β-actin). Mice in four groups (0 h, 24 h, 3 days, and 5 days) were euthanized on the same day to prepare hippocampal samples of all the section slices. Hippocampal DAAO expression was significantly increased by 0.38 times, 0.64 times, and 1.1 times at 24 h (*p* = 0.08), 3 days (*p* = 0.008), and 5 days (*p* = 0.0003), respectively, following MCAO surgery compared to the sham group, as calculated by one-way ANOVA with post hoc Student–Newman–Keuls test (F(3, 16) = 13.4, *p* = 0.0001, Figure 2A).

Double-labeled nestin/DAAO as well as GFAP/DAAO/DAPI immunostaining were applied to study DAAO expression in hippocampal astrocytes at different timepoints after ischemic insult. Of note, nestin is a much more specific marker for reactive astrocytes, and the quantification of nestin/DAAO colocalization is critical to reveal expression of hippocampal DAAO in activated astrocytes. Double-labeled nestin/DAAO immunocolocalization was quantified at four timepoints (0 h, 24 h, 3 days, and 5 days) following MCAO surgery with a computer-assisted image analysis program (F(3, 16) = 204.5, *p* < 0.0001; 0.74 ± 0.06%, 1.61 ± 0.09%, 7.12 ± 0.43%, and 8.88 ± 0.35%, respectively). As shown in Figure 2B–N, the intensity of nestin/DAAO was strikingly increased by more than 11.0 times on day 5 after MCAO onset (*p* = 0.0001, as calculated by one-way ANOVA with post hoc Student–Newman–Keuls test).

Hippocampal DAAO activity was also assessed using a fluorometric assay and byproduct H_2_O_2_ detection. DAAO activity was significantly increased over time (KYNA assay: F(3, 36) = 7.196; 6.14 ± 0.39, 7.72 ± 0.58, 9.32 ± 1.00, and 11.08 ± 1.01, respectively; hydrogen peroxide detection: F(3, 36) = 2.242; 105.0 ± 6.3, 125.7 ± 6.9, 136.4 ± 12.96, and 141.4 ± 15.33, respectively), with maximum increases of 80.6% for the fluorometric assay and 34.6% for H_2_O_2_ detection on day 5 after ischemic insult (*p* = 0.0002 and *p* = 0.042, respectively, calculated by one-way ANOVA with post hoc Student–Newman–Keuls test; Figure 2O,P).

### 3.3. Time Courses of Delayed Hippocampal Neuronal Death in Ischemic Stroke

Neurobehavioral deficit scoring was performed first to evaluate MCAO-induced cerebral damage at four timepoints (0 h, 24 h, 3 days, and 5 days) following MCAO surgery (Figure 3A; F (3, 20) = 34.72, *p* < 0.0001; 0.17 ± 0.17, 1.83 ± 0.31, 2.82 ± 0.31, and 3.67 ± 0.21, respectively). Intravenous, twice-daily injections of 4*H*-furo[3,2-b]pyrrole-5-carboxylic acid DAAO inhibitor (SUN, 10 mg/kg) notably inhibited behavioral deficits on day 5 after MCAO onset by 38.0% compared to the model group treated with saline (t = 2.90, *p* = 0.016, as calculated by unpaired two-tailed Student’s *t*-test; Figure 3B). This experiment confirmed previously established data that SUN reduces neurological deficits following MCAO, but it was assessed using a different system for scoring.

The immunostaining colocalization of hippocampal GFAP/DAAO was performed on day 5 in MCAO mice. The results indicate that DAAO was mainly expressed in GFAP-positive astrocytes in the hippocampal sections (Figure 4A–D). In addition, hippocampal immunofluorescence with NEUN and TUNEL staining was executed (Figure 4E–H). The results show that apoptotic neurons are also mainly localized in the hippocampal CA2 area. Thus, the activation of astrocytic DAAO during ischemic stroke might contribute to neuronal death. In the Nissl-staining study, the numbers of normal neurons and injured cells in the hippocampal CA2 area were recorded on day 5 after ischemic stroke (Figure 4I–K). Hippocampal neurons containing Nissl bodies were recorded as normal cells. Pyknotic nuclei with condensed chromatin were defined as injured cells. The number of total neurons in the model group treated with saline was deceased by 29.7% compared to the sham group, which was alleviated by the administration of SUN (10 mg/kg) at an elevation of 17.3% (t = 3.96, *p* = 0.0027, as calculated by unpaired two-tailed Student’s *t*-test; Figure 4L). In particular, in the analysis of injured cells, injured neurons were significantly decreased in the treatment group by 82.7% compared to model rats treated with saline (t = 37.3, *p* = 0.0001, as calculated by unpaired two-tailed Student’s *t*-test; Figure 4M).

### 3.4. Construction of pcDNA3.1(+)-daao in Primary Astrocytes and Apoptotic Test

To further confirm the role of hippocampal DAAO in MCAO-induced delayed neuronal death, pcDNA3.1(+)-*daao*-transfected primary astrocytes were used to mimic the impact of DAAO overexpression on hippocampal neuronal death in an ischemia–reperfusion animal model. RT-qPCR and Western blot technology were applied to evaluate the mRNA and protein levels of DAAO in pcDNA3.1(+)-*daao*-transfected primary astrocytes. DAAO was overexpressed in transfected primary astrocytes compared to both normal astrocytes and blank pcDNA3.1(+)-transfected astrocytes (Figure 5A,B).

FAD is an essential cofactor for the catalytic activity of DAAO; thus, it should be added to all tests of DAAO activity. After incubating D-Ala (0.3 mM or 1.0 mM) and 2.5 μM FAD with pcDNA3.1-transfected astrocytes for 6 h, the ACM was transferred to a 96-well plate seeded with primary neurons (1 × 10^5^) for apoptotic testing. The apoptotic status of the hippocampal neurons was determined using a fluorescent TUNEL assay kit, with the greatest increases of 1.7 times and 2.2 times on day 2 and day 4, respectively, after incubation in ACM containing 1.0 mM D-Ala (*p* = 0.012 and *p* = 0.0034, respectively, calculated by one-way ANOVA with post hoc Student–Newman–Keuls test; Figure 5C–M). TUNEL-positive cells were also increased by 60.3% and 86.4% on days 2 and 4, respectively, after incubation in ACM containing 0.3 mM D-Ala (*p* = 0.013 and *p* = 0.0016, respectively). The levels of H_2_O_2_, as the byproduct of D-Ala, were also measured in pcDNA3.1(+)-*daao*-transfected astrocytes treated with different D-Ala concentrations (0.03 mM, 0.3 mM, and 1.0 mM) after enzymatic reaction for 30 min (Figure 5N). These results imply that DAAO-induced neuronal apoptosis might be associated with byproduct hydrogen peroxide through the enzymatic reaction of the D-Ala substrate.

### 3.5. Involvement of Transient Receptor Potential (TRP) Calcium Channels and H_2_O_2_ in DAAO-Induced Neuronal Apoptosis

ROS (including H_2_O_2_) directly activate TRP calcium channels (such as TRPA1, TRPV1, TRPM2, TRPC5, etc.), promote calcium influx [21,22], aggravate mitochondrial calcium ion overloading [23], and induce neuronal apoptosis [24,25]. Therefore, many drugs have been employed to explore the potential mechanism of ACM-induced neuronal apoptosis. DAAO inhibitor SUN (50 μM), nonselective TRP calcium channel antagonist SKF96365 (10 μM), and active oxygen scavenger PBN (10 μM) were added to transfected astrocytes with substrate D-Ala and incubated for one day. Different ACM types from treatment subgroups showed diverse improvements in cell viability and cytotoxicity in the sequence of PBN > SUN > SKF96365, as calculated using one-way ANOVA with post hoc Student–Newman–Keuls test (Figure 6A,B). The results demonstrate that TRP calcium channels and H_2_O_2_ could be involved in DAAO-induced neuronal apoptosis.

## 4. Discussion

DAAO is traditionally recognized as a glial enzyme localized in the peroxisome of astrocytes and Bergmann glia [26,27]. The results presented in this study confirm that hippocampal DAAO is specifically expressed in reactive astrocytes and only barely in the neurons of ischemia–reperfusion mice. In our earlier study, we found that an ischemic insult strikingly stimulates astrocytes in the cortical peri-infarct area, resulting in hypertrophy or hyperplasia, which results in formation of a cellular barrier, and DAAO was specifically expressed in reactive astrocytes [9]. Cerebral infarctions and neurobehavioral deficits induced by transient MCAO deteriorated over time, particularly on day 5 after surgery, which was related in parallel to a continual increase in DAAO expression. We therefore suppose that there might be a causal relationship between the high expression of DAAO and cerebral damage in ischemia–reperfusion mice. In this study, an identical situation was also demonstrated in hippocampal sections. The intravenous injection of DAAO inhibitor SUN distinctly reversed delayed hippocampal neuronal death in MCAO mice. Thus, it appears that high expression of DAAO might be also related to delayed neuronal death after ischemic insult. We found that the DAAO inhibitor SUN markedly relieves the degradation in behavioral status and delays hippocampal neuronal death in ischemic mice.

In transfected primary astrocytes, the overexpression of DAAO accompanying astrocytic activation is highly cytotoxic due to the byproduct hydrogen peroxide, which is a stable reactive oxygen species, and significant neuronal apoptosis and necrosis occur [28]. DAAO-mediated hyperalgesia, morphine nociceptive tolerance, and neurodegenerative damage are speculated to be associated with hydrogen peroxide [1,2,24]. Our previous results demonstrated that the DAAO inhibitor SUN distinctly diminishes the whole-brain levels of hydrogen peroxide [9]. Thus, we postulated that the DAAO/hydrogen peroxide pathway is at least partly responsible for delayed neuronal death and neurobehavioral deficits. To further explore direct evidence for the relationship between delayed neuronal death and DAAO overexpression, an in vitro DAAO overexpression model was applied by transfecting the target plasmid into primary astrocytes, and ACM was used for incubation with primary neurons to induce apoptosis. Meanwhile, our results indicated that both TRP calcium channels and H_2_O_2_ are involved in DAAO-induced neuronal apoptosis. H_2_O_2_ can cause astrocytic metabolism disorder, induce an increase in active oxygen free radicals, and decrease BDNF expression [10,11]. Hence, it is supposed that the high level of ROS and reduced BDNF in ACM might elicit apoptosis in neuronal cells. ROS (including H_2_O_2_) directly activate TRP calcium channels (such as TRPA1, TRPV1, TRPM2, TRPC5, etc.), promote calcium influx [21,22], and aggravate mitochondrial calcium ion overloading [23], resulting in the downregulation of a variety of synaptic protein expression and neuronal apoptosis [24,25]. In addition, the low levels of BDNF in ACM might inactivate the functions of tyrosine kinase receptor B in cellular survival and synaptic plasticity via phosphoinositide 3-kinase/protein kinase B, mitogen-activated protein kinase, etc. [29,30].

A few limitations of this study need to be reported. We did not set up a sham group as a part of the study design, which is a limitation of this study. Our outcomes showed that DAAO and its substrate D-Ala only promoted neuronal apoptosis in a dose-dependent manner. Whether DAAO overexpression in neuropathological disorders could also exhibit cytotoxicity is undefined. The influence of inhibiting DAAO on learning ability and memory should also be confirmed in further work.

In conclusion, we found that astrocytic activation induced by ischemic insult is concomitant with the increasing expression of DAAO over a time period of several days. In addition, the current outcomes suggest that the high expression of hippocampal DAAO is causally associated with delayed neuronal death at subacute phases of ischemia–reperfusion, at least partly, by enhancing H_2_O_2_ levels with subsequent activation of TRP calcium channels in neurons. This study demonstrated the involvement of DAAO in astrocyte–neuron interactions in neuropathological disorders as well as that the mechanism underlying neuronal apoptosis is induction by astrocytic DAAO. DAAO could thus serve as a therapeutic target in the management of ischemic stroke and other neurodegenerative diseases.

## Figures and Tables

**Figure 1 cells-11-03689-f001:**
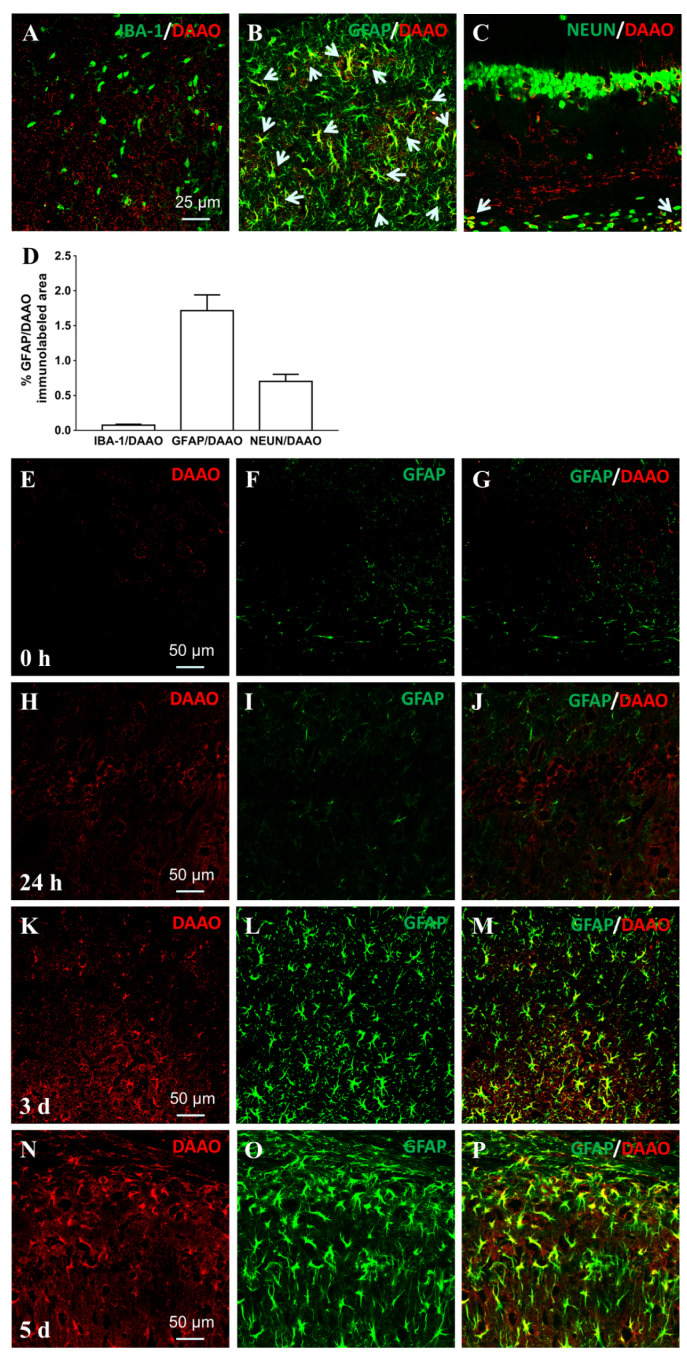
Hippocampal DAAO is mainly expressed in activated astrocytes in ischemic mice induced by transient MCAO. Immunostained DAAO in hippocampal CA2 area is double-labeled with the biomarkers Iba-1 for microglia (**A**), GFAP for astrocytes (**B**), and NeuN for neurons (**C**) on day 3 after ischemic surgery (*n* = 3, 0.2 mm × 0.2 mm). Arrows indicate the colocalization of DAAO (red) and cell-specific biomarkers (green). The colocalized area fractions of DAAO with three cellular markers were quantified (**D**). Data are shown as means ± SEM. Double-labeled GFAP/DAAO immunostaining was also colocalized and quantified at four timepoints (0 h, 24 h, 3 days, and 5 days) following MCAO surgery (**E**–**P**).

**Figure 2 cells-11-03689-f002:**
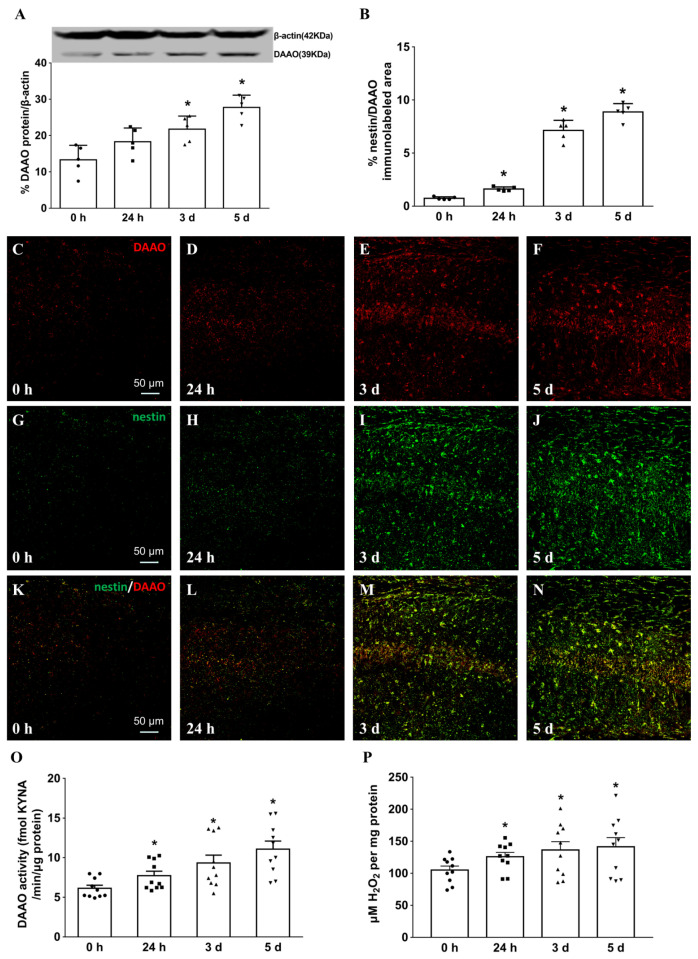
Time courses of hippocampal DAAO expression and related activity after acute ischemic insult. Western blot (**A**), colocalized nestin/DAAO immunostaining (**B**–**N**), and DAAO activity assays (**O**,**P**) were employed to study time courses of hippocampal DAAO after surgery (0 h, 24 h, 3 days, and 5 days). Hippocampal homogenate supernatant in 0.4 M Tris-buffer (pH 8.5) was used as the DAAO source for enzymatic activity assays. The colocalized areas of hippocampal nestin/DAAO immunostaining were quantified using ImageJ. Data are shown as means ± SEM (*n* = 4–5 in each group). * denotes statistical significance (*p* < 0.05) compared with mice at the timepoint of 0 h, as analyzed by one-way ANOVA with post hoc Student–Newman–Keuls test.

**Figure 3 cells-11-03689-f003:**
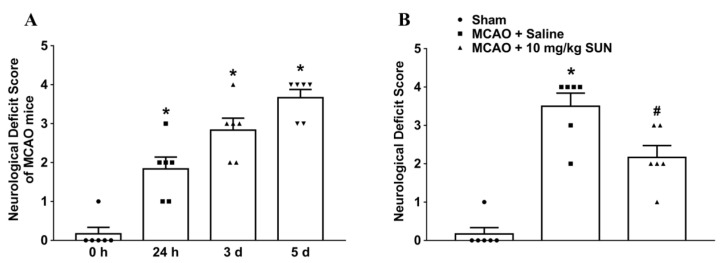
Time course of behavioral deficits after MCAO surgery. (**A**) Neurobehavioral deficit score was adopted to evaluate MCAO-induced cerebral damage. (**B**) Intravenous, twice-daily injections of DAAO inhibitor SUN (10 mg/kg) for 5 days notably inhibited behavioral deficits on day 5 after MCAO onset. Data are expressed as means ± SEM (*n* = 6). * denotes statistical significance (*p* < 0.05) compared with mice at the timepoint of 0 h or in the sham group; # denotes significant difference from model group treated with saline (*p* < 0.05) calculated by one-way ANOVA with post hoc Student–Newman–Keuls test or unpaired two-tailed Student’s *t*-test (nonparametric statistics for neurological scores).

**Figure 4 cells-11-03689-f004:**
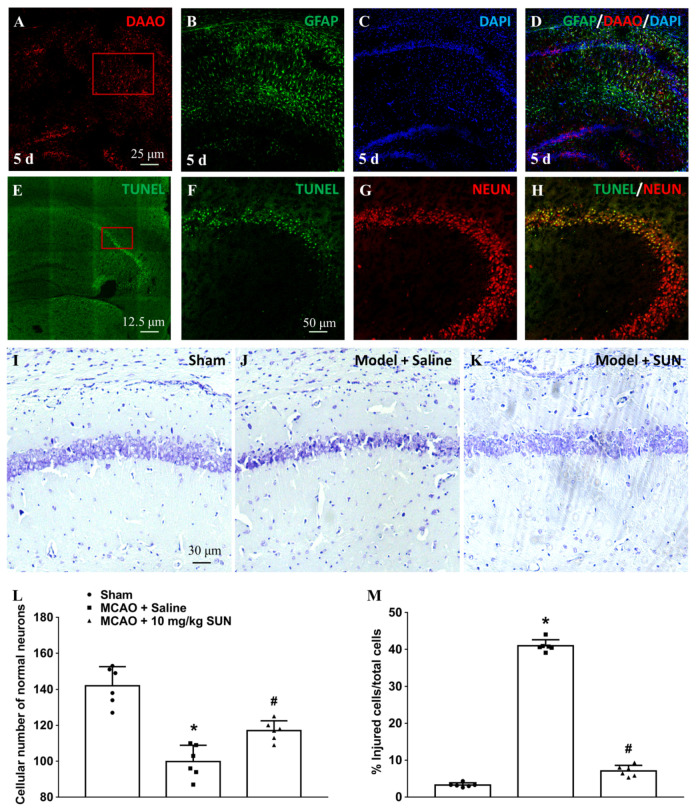
Effect of DAAO inhibitor SUN in preventing delayed hippocampal neuronal death after MCAO surgery. (**A**–**D**) Immunostaining colocalization of hippocampal GFAP/DAAO was performed on day 5 in MCAO mice. (**E**–**H**) Hippocampal immunofluorescence NEUN and TUNEL staining were executed on day 5 in MCAO mice. The red rectangle represents the hippocampal CA2 region. (**I**–**M**) Numbers of normal neurons and injured cells were identified by Nissl staining in hippocampal CA2 area on day 5 after ischemic strokes. Hippocampal neurons containing Nissl bodies were recorded as normal cells. Pyknotic nuclei with condensed chromatin were defined as injured cells. Data are expressed as means ± SEM (*n* = 6). * denotes statistical significance (*p* < 0.05) compared with sham group; # denotes significant difference from the model group treated with saline (*p* < 0.05) calculated by unpaired two-tailed Student’s *t*-test.

**Figure 5 cells-11-03689-f005:**
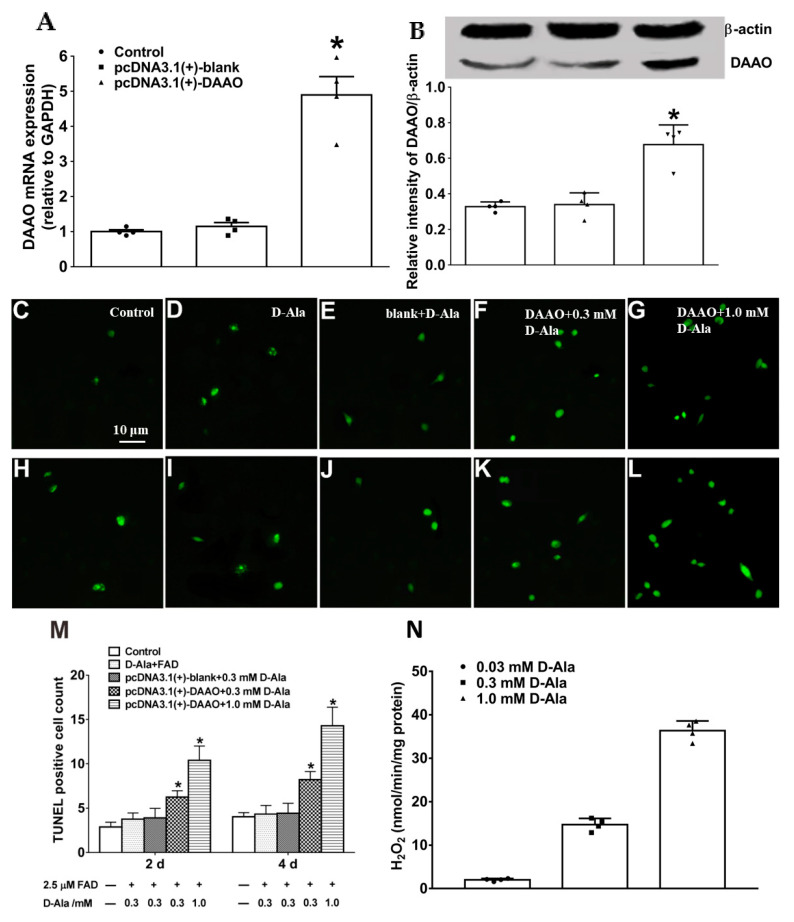
DAAO overexpression promotes hippocampal neuronal death. (**A**,**B**) Evaluation of pcDNA3.1(+)-*daao* transfection and DAAO expression at both mRNA and protein levels. The transfected astrocytes were collected using a cell scraper. The concentration of total protein extracted from cells with lysis solution was determined using the bicinchoninic acid protein method. (**C**–**L**) Apoptotic neurons were detected using a fluorescent TUNEL assay kit at different timepoints (2 d: (**C**–**G**) and 4 d: (**H**–**L**)). (**M**) Cell apoptosis was quantitatively analyzed with ImageJ software. (**N**) The H_2_O_2_ level was measured as the byproduct of different concentrations of D-Ala in pcDNA3.1(+)-*daao*-transfected astrocytes after enzymatic reaction for 30 min. Data are expressed as means ± SEM (*n* = 4). * denotes statistical significance (*p* < 0.05) compared with pcDNA3.1(+)-blank+0.3 mM D-Ala group, as calculated by one-way ANOVA and post hoc Student–Newman–Keuls tests (**M**) or unpaired two-tailed Student’s *t*-test (**A**,**B**).

**Figure 6 cells-11-03689-f006:**
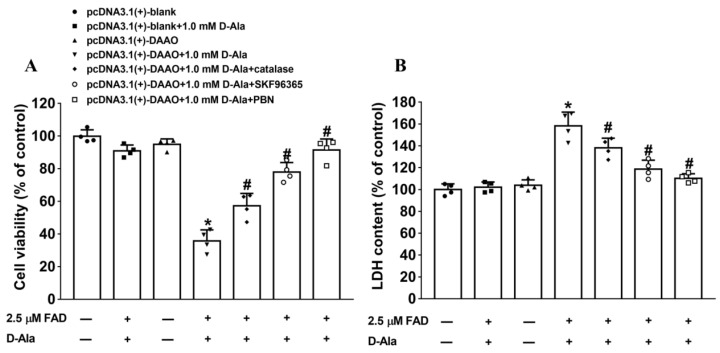
DAAO overexpression inhibits cell activity and induces cytotoxicity. (**A**) Cell activity was measured with a CCK-8 assay kit. (**B**) Cytotoxicity was assessed based on an LDH leakage assay. DAAO inhibitor SUN (50 μM), nonselective TRP calcium channel antagonist SKF96365 (10 μM), and active oxygen scavenger PBN (10 μM) were applied to evaluate whether TRP calcium channels and H_2_O_2_ are involved in DAAO-induced neuronal apoptosis. * denotes statistical significance (*p* < 0.05) compared with pcDNA3.1(+)-blank+1.0 mM D-Ala group; # denotes significant difference from model group treated with saline (*p* < 0.05) calculated by one-way ANOVA and post hoc Student–Newman–Keuls test.

## Data Availability

All the data generated or analyzed during this study are included in the published article.

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
