# Peer review of "Involvement of DAAO Overexpression in Delayed Hippocampal Neuronal Death"

_cells, 2022, doi:10.3390/cells11223689_

Round 1
Reviewer 1 Report
I have read the manuscript of Liu et al. entitled "Involvement of DAAO overexpression in delayed hippocampal neuronal death". The manuscript reports a technically and scientifically sound study that may be of interest for other scientists making research on the role of D-amino acid oxidase in physiology and pathology of the central nervous system. However, the manuscript is not yet suitable for publication in its present form because of poor language and editing issues. Furthermore, the Authors need to elaborate more on the description of some methodological aspects, on the reporting of results, as well as on the explanation of the research aims and conclusions.
Please, consider my following comments:
1) The manuscript needs extensive language editing! In the present form, some parts of the manuscript are hard to understand because of wrong phrasing and grammar errors. There are some unusually phrased sentences, such as:
"The mouse death only started on Day 4-5 after surgery with death rate of about 20%."
"The cerebral sections were baked in air incubator..."
"After incubating D-Ala (0.3 mM or 1.0 mM) and 2.5 μM FAD with pcDNA3.1-transfected astrocytes for 6 hours, the ACM would be shifted into culture plates with primary neurons seeded for the following studies."
"The intravenous injection of DAAO inhibitor SUN distinctly reversed delayed hippocampal neuronal death by RT-qPCR technology to test NeuN mRNA on Day 5 after MCAO onset."
2) The order of subsections in the Methods section is illogical. It would be more intuitive if the methodological descriptions of in vitro and in vivo experiments were separated better by starting with the in vivo methods and continuing thereafter with in vitro methods. The Authors could follow the order of the results section.
3) It is very hard to catch the main research questions as well as the main message of the study.
The Introduction section is insufficient for the purpose of introducing the general research area to the reader. The Authors should write more about previous findings on the role and properties of DAAO enzyme. Furthermore, the Authors should better define the aims and the scientific question because now it is not really clear.
The Authors should also elaborate more on the Discussion, especially by emphasizing better the conclusions and significance of the findings in the light of future studies and clinical relevance.
4) Much information are missing in the description of the methodologies and experimental design. Some examples for missing details:
- the supplier of some chemicals/reagents
- method for calculation of enzymatic activity in the kynurenic acid assay
- exact dilution of sample supernatants in the H2O2 assay ("moderately diluted")
- it should be clarified somewhere in the Methods (preferably early in the section) what was the design and time-course of the in vivo study (groups, survival times, pharmacological treatments), and that all of the described assays were conducted on the samples from this study (or if not all groups/survival times were analyzed with a given assay, then emphasize which samples were tested). It is very confusing that it is not specified in certain subsections whether the given assay was conducted on in vivo or in vitro samples.
5) The reporting of the results is not systematic. In the text, the results are selectively mentioned, and the text includes almost no numeric data (e.g., mean+/-SEM of a given group). There are some sentences that would better fit into the Methods section. Some kind of quantification is needed for all of the histological experiments (now, some immunofluorescent labelings were quantified and compared statistically, others were not).
6) The control group(s) is/are not clearly indicated in the Results section. Is "0 h" group in some figures and comparisons the sham operated group, or is it an MCAO-treated group that was sacrified immediately after surgery? Authors should clearly define the control groups in the presentation of the results. Furthermore, the sham operated control is lacking in the reporting of some experiments, e.g., in subsection 3.1.
7) An important conceptual issue is that the in vivo and in vitro experiments are not tightly connected to each other. The Authors should better emphasize in the aims or in the interpretation of the results, what is the link between functional and pharmacological studies in MCAO mice and in vitro measurements in astrocyte cell cultures.
8) The Authors confirmed earlier findings that DAAO is mainly expressed by astrocytes in the CNS. However, the present in vivo studies focused on the MCAO mouse model of cerebral circulatory diseases but it was not compared in the study whether there is any difference in the expression profiles of DAAO in MCAO and in healthy animals. To ensure coherence of the paper, a comparison between sham and MCAO mice would be of great importance.
9) On Figure 2G-H, the labeling of GFAP (astrocytes) seems unusual to me. Astrocytes are quite abundant even in healthy brain samples, however, on these photomicrographs a relatively few GFAP+ cells can be seen. What may be the reason of this unusual pattern? Again, a proper control photomicrograph (if "0 h" is not the sham group, see above) would help in the evaluation of these histological patterns.
10) Figure 5C-G and H-L should be clearly labeled: now, it is not clear which image corresponds to which group/treatment/time-point. Please, clarify.
Author Response
Please, consider my following comments:
1) The manuscript needs extensive language editing! In the present form, some parts of the manuscript are hard to understand because of wrong phrasing and grammar errors. There are some unusually phrased sentences, such as:
"The mouse death only started on Day 4-5 after surgery with death rate of about 20%."
"The cerebral sections were baked in air incubator..."
"After incubating D-Ala (0.3 mM or 1.0 mM) and 2.5 μM FAD with pcDNA3.1-transfected astrocytes for 6 hours, the ACM would be shifted into culture plates with primary neurons seeded for the following studies."
"The intravenous injection of DAAO inhibitor SUN distinctly reversed delayed hippocampal neuronal death by RT-qPCR technology to test NeuN mRNA on Day 5 after MCAO onset."
Reply: Many thanks for your suggestion, it was revised accordingly.
2) The order of subsections in the Methods section is illogical. It would be more intuitive if the methodological descriptions of in vitro and in vivo experiments were separated better by starting with the in vivo methods and continuing thereafter with in vitro methods. The Authors could follow the order of the results section.
Reply: The order of subsections in methods section was restructured accordingly.
3) It is very hard to catch the main research questions as well as the main message of the study.
The Introduction section is insufficient for the purpose of introducing the general research area to the reader. The Authors should write more about previous findings on the role and properties of DAAO enzyme. Furthermore, the Authors should better define the aims and the scientific question because now it is not really clear.
The Authors should also elaborate more on the Discussion, especially by emphasizing better the conclusions and significance of the findings in the light of future studies and clinical relevance.
Reply: The additional information was added in introduction section and conclusion section.
4) Much information are missing in the description of the methodologies and experimental design. Some examples for missing details:
- the supplier of some chemicals/reagents
- method for calculation of enzymatic activity in the kynurenic acid assay
- exact dilution of sample supernatants in the H2O2 assay ("moderately diluted")
- it should be clarified somewhere in the Methods (preferably early in the section) what was the design and time-course of the in vivo study (groups, survival times, pharmacological treatments), and that all of the described assays were conducted on the samples from this study (or if not all groups/survival times were analyzed with a given assay, then emphasize which samples were tested). It is very confusing that it is not specified in certain subsections whether the given assay was conducted on in vivo or in vitro samples.
Reply: The additional information was added accordingly in method subsections.
5) The reporting of the results is not systematic. In the text, the results are selectively mentioned, and the text includes almost no numeric data (e.g., mean+/-SEM of a given group). There are some sentences that would better fit into the Methods section. Some kind of quantification is needed for all of the histological experiments (now, some immunofluorescent labelings were quantified and compared statistically, others were not).
Reply: The additional information was added accordingly.
6) The control group(s) is/are not clearly indicated in the Results section. Is "0 h" group in some figures and comparisons the sham operated group, or is it an MCAO-treated group that was sacrified immediately after surgery? Authors should clearly define the control groups in the presentation of the results. Furthermore, the sham operated control is lacking in the reporting of some experiments, e.g., in subsection 3.1.
Reply: The definitions of control group and sham group were in the end of “The mouse MCAO model” subsection. The sham group was mainly operated to confirm the impact of surgical wound itself but not ischemia/reperfusion to the neurobehavioral score and hippocampal neuronal death in efficacy study. The data in subsection 3.3 and 3.4 showed that surgical wound did not lead to the obvious neuropathological change. For “0 h” group, normal mouse was selected. In subsection 3.2, the pathological changes of DAAO in different time points of ischemic stroke were studied, and “0 h” group was applied as control actually. At last, we indeed did not set up the sham operated control in subsection 3.2, which was a limitation of this study.
7) An important conceptual issue is that the in vivo and in vitro experiments are not tightly connected to each other. The Authors should better emphasize in the aims or in the interpretation of the results, what is the link between functional and pharmacological studies in MCAO mice and in vitro measurements in astrocyte cell cultures.
Reply: The aims of in vivo and in vitro studies were emphasized in the end of introduction and subsections of “2.3. The mouse MCAO model” and “2.10 Construction of pcDNA3.1(+)-DAAO transfected primary astrocyte”.
8) The Authors confirmed earlier findings that DAAO is mainly expressed by astrocytes in the CNS. However, the present in vivo studies focused on the MCAO mouse model of cerebral circulatory diseases but it was not compared in the study whether there is any difference in the expression profiles of DAAO in MCAO and in healthy animals. To ensure coherence of the paper, a comparison between sham and MCAO mice would be of great importance.
Reply: In this study, the change of DAAO expression was detected according to the time point after surgery. The normal mice were defined as MCAO mice on Day 0. We indeed did not make a comparison between sham and MCAO mice in these immunofluorescent experiments, which was a limitation of this study.
9) On Figure 2G-H, the labeling of GFAP (astrocytes) seems unusual to me. Astrocytes are quite abundant even in healthy brain samples, however, on these photomicrographs a relatively few GFAP+ cells can be seen. What may be the reason of this unusual pattern? Again, a proper control photomicrograph (if "0 h" is not the sham group, see above) would help in the evaluation of these histological patterns.
Reply: Thanks for your suggestion. Astrocytes indeed are abundant in the healthy brain. However, the GFAP is expressed low in physiological condition. Both expressions of GFAP and nestin proteins are enhanced in activated astrocytes, followed with the pathological duration as below.
(References: Fig. 2 in article with PMID: 22940431).
(Our data from Figure 2 H-J)
10) Figure 5C-G and H-L should be clearly labeled: now, it is not clear which image corresponds to which group/treatment/time-point. Please, clarify.
Reply: Many thanks for your suggestion, it was revised accordingly.

Reviewer 2 Report
Review of Liu et al: Involvement of DAAO overexpression in delayed hippocampanl neuronal death
This manuscript addresses the role of D-amino acid oxidase (DAAO) in astrocytes, and its potential role in mediating neuronal death in a model of ischaemic stroke. The authors report that DAAO is upregulated following middle cerebral artery occlusion (MCAO), and that this expression is restricted to astrocytes, correlating with increased H2O2 levels in the hippocampus. They further demonstrate that inhibition of DAAO activity was able to protect mice against the neurological deficits associated with the model, and that it reduced the loss or injury of neurons within the hippocampus. Next, cultured astrocytes were transfected to overexpress DAAO, exposed to D-Ala as a DAAO substrate, and the harvested medium was subsequently used to treat cultured neurons and induce cell death. They then used inhibitors of either DAAO, TRP channels, or a hydrogen peroxide scavenger to characterize the pathways involved in neuronal death.
General comments:
The main concern with the paper is that it is not made clear exactly what is already known in the field, what the specific objectives of this study are, and what new knowledge has been obtained as a result (or if they have just confirmed previously known findings). In fact, it appears that Figures 1, 2 and 3 are a repeat of data already published by the group (Liu et al., 2019. Br J Pharmacol). The writing also needs to be checked by a native English speaker since there are numerous mistakes, and sometimes the meaning is not clear. Furthermore, the logic of some of the experiments is not made clear.
Specific comments:
Abstract
- It is not enough to say ‘TRP channels and H2O2 might be involved – please either give a reason for this, or leave out.
- What does ‘latent modality’ mean exactly, a target for delayed stroke treatment?
Introduction
- This is far too short with not enough information given to know what the study is about. What does DAAO do specifically (how does it affect the physiological and pathophysiological functions that are listed – if this is not known, then please explain since this might help the reader to understand the relevance of the experiments performed here).
- It states: ‘our previous report disclosed the … expression … in activated astrocytes’ but then says that the cell specificity is not known. Please elaborate – since it appears contradictory. Is it known in other brain regions, but not hippocampus?
- Is not clear what is new and different in this study to previous work that has been done. Please give a clear explanation about which information exactly is not already known and what the aims of this study are.
- It would help the rest of the paper make sense if it is explained already here that a byproduct of D-amino acid degradation by DAAO is H2O2 which can be toxic.
Methods:
- Many useful details are missing such as companies and concentrations of reagents. For example, in section 2.3, no companies are given for the basic chemicals used, and the concentration of B27 in the neuronal media is missing.
- Section 2.7: More information is needed about the preparation of cerebral sections – which tissue was used? Was it perfused, was it fixed, how was it prepared for frozen sectioning, how thick were the sections?
- Section 2.8: Which samples were used for Western blotting? Presumably the hippocampus was dissected and then lysed? How were lysates prepared (what solutions – did they contain protease inhibitors, etc)? What concentrations/protein amounts were used for Western blotting?
- Section 2.9: What was the concentration of the hippocampal homogenate used for the assay (not just the volume). How were homogenates generated – presumably following dissection of hippocampi, they were added to the Tris buffer and then homogenized – i.e. generated differently to those used for Western blotting? ‘Reactive solution’ should be rephrased since confusing. For the H2O2 assay – were homogenates generated in exactly the same way as for the DAAO activity assay? ‘Moderately diluted’ supernatant doesn’t mean anything – either explain so someone else can perform the experiment, or just state ‘prepared according to the kit manufacturer’s instructions’.
- Section 2.10: Please clarify that the coronal sections were generated from hippocampus. Again, was it perfused, was it fixed, it was presumably dehydrated and then paraffin embedded, what thickness were sections cut at?
- Section 2.11. DAAO cDNA was amplified from what source? Which promoter was used to drive protein expression? When were astrocytes transfected – at time of seeding, or after being established for a certain number of days? They were then presumably cultured for a further 2 days following application of the transfection mix.
- Section 2.12: Please explain that ACM means astrocyte conditioned media. Was it not supplemented with B27 or Gln to support neuronal growth? Also, please explain briefly what the CCK-8 assay kit is measuring. Were the culture plates coated with anything to aid neuronal adhesion?
Results:
- Figure 1: Which region of the hippocampus is being shown? What do the arrows indicate?
- Section 3.2 – increases in DAAO expression are given as 0.38, 0.54 and 1.1 times. This is confusing – a 1 times increase would normally denote no change. It would be best to give the values as percentages of the control. I wouldn’t have said that the nestin/DAAO quantification was ‘critical’ to reveal the expression, but that it serves as an aid? Please mention in the text what was quantified exactly – i.e. the ‘area’ of colocalisation.
- Figure 2: What does sham refer to – sham-treated samples from the respective time-points which are not shown here, or the 0 hour time-point that is shown? Please make this clear, and if necessary, the sham Western blots used for comparison should be shown too. The legend should be expanded to explain that hippocampal lysates were used to generate the data in panels K and L.
- Figure 3. This figure is presumably a confirmation of data previously published by the group, as explained elsewhere in the text? This should be clarified here, and repeated data would probably not merit their own dedicated figure?
- Figure 4: ‘staining was executed successively’ This phrase (which also appears in both the legend and explaining text) is unclear – were the stainings performed in succession or was it successful, in which case it is not needed since only successful stainings would be included in the manuscript. Please show sham images for the TUNEL staining. I would be interested to know if treatment with SUN reduces the TUNEL staining? Higher magnification images of Panels I-K would help to identify the pyknotic nuclei.
- Section 3.5: This section needs more clarification since the logic of the experiments is not very clear. The authors explain that DAAO is expressed in peroxisomes, but is DAAO also secreted? If not, how is the addition of SUN to the conditioned media expected to work? I imagine that the experiment is designed to assess the neurotoxic effects of released H2O2, as confirmed by addition of PBN. Direct measurement of H2O2 in the conditioned media using the previously used H2O2 assay would be helpful in this regard, as would addition of H2O2 alone at the concentrations measured. The effect of PBN administration in vivo would also be interesting.
- Section 3.5 continued: Please explain somewhere that FAD is an essential cofactor for DAAO. The values and statistics are given for 1.0mM D-Ala, but are missing for 0.3mM D-Ala (which are also significant). Please explain the logic for investigating the role of TRP channels here since it just appears without explanation.
Discussion
- The sentence starting ‘It was identical with our work …’ is not clear whether it refers to the data shown in this manuscript or that which was previously published. The authors should make it clear what was previously known about DAAO and stroke, and what is novel to this study.
- Much of the information included in the discussion would be well suited to the Introduction such as details regarding the role of DAAO overexpression in astrocytes being toxic due to production of hydrogen peroxide.
General typos:
Section 2.7: Tissue had antigens ‘retrieved’ rather than repaired? Triton X-100 was used to ‘permeablise’ the tissue rather than as ‘sealing fluid’?
Section 2.10: Misspelled ‘investigator’
Section 3.2: By ‘Synaptic extension’, I presume the authors mean ‘processes’ and not synapses. Same for the discussion: ‘‘resulting in hypertrophy or hyperplasia with synapses’ presumably refers to processes, too.
Discussion: ‘parallelly’ should read ‘in parallel’
Author Response
Specific comments:
Abstract
- It is not enough to say ‘TRP channels and H2O2 might be involved – please either give a reason for this, or leave out.
- What does ‘latent modality’ mean exactly, a target for delayed stroke treatment?
Reply: It was revised in abstract and other sections accordingly.
Introduction
- This is far too short with not enough information given to know what the study is about. What does DAAO do specifically (how does it affect the physiological and pathophysiological functions that are listed – if this is not known, then please explain since this might help the reader to understand the relevance of the experiments performed here).
Reply: The additional information was explained in introduction section.
- It states: ‘our previous report disclosed the … expression … in activated astrocytes’ but then says that the cell specificity is not known. Please elaborate – since it appears contradictory. Is it known in other brain regions, but not hippocampus?
Reply: Yes you are right. The cell specificity of cortical DAAO was reported in our previous study. We only stated that it was necessary to clarify this situation in hippocampus as well.
- Is not clear what is new and different in this study to previous work that has been done. Please give a clear explanation about which information exactly is not already known and what the aims of this study are.
Reply: The aims of in vivo and in vitro studies were emphasized in the end of introduction and subsections of “2.3. The mouse MCAO model” and “2.10 Construction of pcDNA3.1(+)-DAAO transfected primary astrocyte”. The underlying mechanism of DAAO-induced neuronal death was explained by astrocyte-neuron interactions.
- It would help the rest of the paper make sense if it is explained already here that a byproduct of D-amino acid degradation by DAAO is H2O2 which can be toxic.
Reply: Many thanks for your suggestion. The related information was added accordingly.
Methods:
- Many useful details are missing such as companies and concentrations of reagents. For example, in section 2.3, no companies are given for the basic chemicals used, and the concentration of B27 in the neuronal media is missing.
Reply: The related information was added accordingly.
- Section 2.7: More information is needed about the preparation of cerebral sections – which tissue was used? Was it perfused, was it fixed, how was it prepared for frozen sectioning, how thick were the sections?
Reply: The related information was added accordingly.
- Section 2.8: Which samples were used for Western blotting? Presumably the hippocampus was dissected and then lysed? How were lysates prepared (what solutions – did they contain protease inhibitors, etc)? What concentrations/protein amounts were used for Western blotting?
Reply: The related information was added accordingly.
- Section 2.9: What was the concentration of the hippocampal homogenate used for the assay (not just the volume). How were homogenates generated – presumably following dissection of hippocampi, they were added to the Tris buffer and then homogenized – i.e. generated differently to those used for Western blotting? ‘Reactive solution’ should be rephrased since confusing. For the H2O2 assay – were homogenates generated in exactly the same way as for the DAAO activity assay? ‘Moderately diluted’ supernatant doesn’t mean anything – either explain so someone else can perform the experiment, or just state ‘prepared according to the kit manufacturer’s instructions’.
Reply: Yes, you are right, the hippocampal homogenate generated for DAAO activity assay was different from that for Western blot assay. The related information was revised accordingly.
- Section 2.10: Please clarify that the coronal sections were generated from hippocampus. Again, was it perfused, was it fixed, it was presumably dehydrated and then paraffin embedded, what thickness were sections cut at?
Reply: It was explained accordingly.
- Section 2.11. DAAO cDNA was amplified from what source? Which promoter was used to drive protein expression? When were astrocytes transfected – at time of seeding, or after being established for a certain number of days? They were then presumably cultured for a further 2 days following application of the transfection mix.
Reply: It was explained accordingly.
- Section 2.12: Please explain that ACM means astrocyte conditioned media. Was it not supplemented with B27 or Gln to support neuronal growth? Also, please explain briefly what the CCK-8 assay kit is measuring. Were the culture plates coated with anything to aid neuronal adhesion?
Reply: The ACM was explained in the text when shown for the first time and Abbreviations section. The supplementary of B27 and Gln was also added into ACM to support neuronal growth. The measurement of CCK-8 assay kit was explained. The culture plates were coated with poly-L-lysine to aid neuronal adhesion.
- Figure 1: Which region of the hippocampus is being shown? What do the arrows indicate?
Reply: It was explained accordingly.
- Section 3.2 – increases in DAAO expression are given as 0.38, 0.54 and 1.1 times. This is confusing – a 1 times increase would normally denote no change. It would be best to give the values as percentages of the control. I wouldn’t have said that the nestin/DAAO quantification was ‘critical’ to reveal the expression, but that it serves as an aid? Please mention in the text what was quantified exactly – i.e. the ‘area’ of colocalisation.
Reply: It was added accordingly.
- Figure 2: What does sham refer to – sham-treated samples from the respective time-points which are not shown here, or the 0 hour time-point that is shown? Please make this clear, and if necessary, the sham Western blots used for comparison should be shown too. The legend should be expanded to explain that hippocampal lysates were used to generate the data in panels K and L.
Reply: The definition of each subgroup was explained in 2.3 subsection titled as “The mouse MCAO model”. We indeed did not set up the sham operated control in subsection 3.2, which was a limitation of this study. Hippocampal lysate was explained in the legend of figure 2 for panels K and L.
- Figure 3. This figure is presumably a confirmation of data previously published by the group, as explained elsewhere in the text? This should be clarified here, and repeated data would probably not merit their own dedicated figure?
Reply: The neurobehavioral deficit was assessed by Clark score in our previous study. In this work, Zea-Longa neurological grading scale was applied for the first time.
- Figure 4: ‘staining was executed successively’ This phrase (which also appears in both the legend and explaining text) is unclear – were the stainings performed in succession or was it successful, in which case it is not needed since only successful stainings would be included in the manuscript. Please show sham images for the TUNEL staining. I would be interested to know if treatment with SUN reduces the TUNEL staining? Higher magnification images of Panels I-K would help to identify the pyknotic nuclei.
Reply: ‘staining was executed successively’ This phrase means that the section was stained with anti- NEUN antibody first, and subsequently performed with TUNEL staining. Thus the fluorescent NEUN/TUNEL could be shown in one section (as shown in Figure 4 F-H). In our former study, it was proved that SUN inhibited the ischemic damage by TTC staining (Liu H, et al. British journal of pharmacology 2019, 176 (17), 3336-3349.). In this study, we proved its inhibitory effect on hippocampal neuronal death by Nissl staining. The magnification of images in panels I-K was amplified accordingly.
- Section 3.5: This section needs more clarification since the logic of the experiments is not very clear. The authors explain that DAAO is expressed in peroxisomes, but is DAAO also secreted? If not, how is the addition of SUN to the conditioned media expected to work? I imagine that the experiment is designed to assess the neurotoxic effects of released H2O2, as confirmed by addition of PBN. Direct measurement of H2O2 in the conditioned media using the previously used H2O2 assay would be helpful in this regard, as would addition of H2O2 alone at the concentrations measured. The effect of PBN administration in vivo would also be interesting.
Reply: Many thanks for your comments. The molecular weight of compound SUN was 151 Da and may play functions in cells by simple diffusion. We indeed detected the H2O2 level in conditioned medium and the related image was inserted as Figure 5N.
- Section 3.5 continued: Please explain somewhere that FAD is an essential cofactor for DAAO. The values and statistics are given for 1.0mM D-Ala, but are missing for 0.3mM D-Ala (which are also significant). Please explain the logic for investigating the role of TRP channels here since it just appears without explanation.
Reply: Many thanks for your comments. The essential role of FAD was explained in “3.4 subsection” as a key coenzyme for catalytic activity of DAAO. The related data about 0.3 mM D-Ala was added as well. The role of TRP channels was explained in “3.5 subsection” accordingly.
Discussion
- The sentence starting ‘It was identical with our work …’ is not clear whether it refers to the data shown in this manuscript or that which was previously published. The authors should make it clear what was previously known about DAAO and stroke, and what is novel to this study.
Reply: The related information was added in the last paragraph of introduction section.
- Much of the information included in the discussion would be well suited to the Introduction such as details regarding the role of DAAO overexpression in astrocytes being toxic due to production of hydrogen peroxide.
Reply: Many thanks for your suggestion. The related information was added in “introduction” section.
General typos:
Section 2.7: Tissue had antigens ‘retrieved’ rather than repaired? Triton X-100 was used to ‘permeablise’ the tissue rather than as ‘sealing fluid’?
Reply: Many thanks for your suggestion. The word “retrieve” was used to replace “repair”. We agree that Triton X-100 was used to increase permeability of cerebral section, and in other hand, “sealing fluid” refer to “10% goat serum” as it is the main component used to seal tissue antigens.
Section 2.10: Misspelled ‘investigator’
Reply: It was revised accordingly.
Section 3.2: By ‘Synaptic extension’, I presume the authors mean ‘processes’ and not synapses. Same for the discussion: ‘‘resulting in hypertrophy or hyperplasia with synapses’ presumably refers to processes, too.
Reply: For its potential misunderstanding, the related words were deleted.
Discussion: ‘parallelly’ should read ‘in parallel’
Reply: It was revised accordingly.

Reviewer 3 Report
The authors have investigated the role of DAAO in cerebral ischemia/reperfusion-induced hippocampal cell death. Although, the manuscript is well written, there are many areas which needs to be improved for better readability and understanding.
1) In Figure 1, please provide information like animal number, area (mm2) analyzed, quantification of co-immunolabeling.
2) The Kda for DAAO is described as 28KDa https://www.novusbio.com/PDFs/NBP1-84305.pdf. In Figure 2, it is described as 39Kda. Please check.
3) Either use boxplot or histograms. Keep it consistent.
4) In Figure 1, write “NeuN for neurons (C)” instead of “NeuN (C) for neurons”.
5) Insert each data point in each bar in the graphs.
6) “Tick mark” should be outside of the y-axis. Keep it consistent too.
7) The numbers are too close to the y-axis. Keep it consistent too.
8) Either use tick marks or delete from the x-axis but keep it consistent.
9) In Figure 3B, ANOVA should be used to analyze the data instead of t-test.
10) Show double or triple co-labeling in an additional high magnification figure.
11) Also, there is no t- or F-value mentioned in the text.
Author Response
The authors have investigated the role of DAAO in cerebral ischemia/reperfusion-induced hippocampal cell death. Although, the manuscript is well written, there are many areas which needs to be improved for better readability and understanding.
1) In Figure 1, please provide information like animal number, area (mm2) analyzed, quantification of co-immunolabeling.
Reply: The related information was added accordingly.
2) The Kda for DAAO is described as 28KDa https://www.novusbio.com/PDFs/NBP1-84305.pdf. In Figure 2, it is described as 39Kda. Please check.
Reply: The molecular weight of DAAO protein is 37 kDa. The position of grey band may be different due to distinct species. It was mainly positive at the position of 35 kDa in cerebellum and 39 kDa in cerebrum or spinal cord in Swiss mouse according to our former experience. (Reference: Jin-Lu Huang, Amino Acids (2012) 43:1905–1918).
3) Either use boxplot or histograms. Keep it consistent.
Reply: There are some conventional rules for the usage of boxplot or histograms. People commonly use boxplot to show the data with non-normal distribution and do the analysis by a non-parametric equivalent.
4) In Figure 1, write “NeuN for neurons (C)” instead of “NeuN (C) for neurons”.
Reply: Many thanks for your suggestion. It was revised.
5) Insert each data point in each bar in the graphs.
Reply: For the conciseness of images as one whole, rare literature would insert each data point for each bar in the graphs. . Instead all data for each graph were enclosed as supplementary file.
6) “Tick mark” should be outside of the y-axis. Keep it consistent too.
Reply: It was revised accordingly.
7) The numbers are too close to the y-axis. Keep it consistent too.
Reply: It was revised accordingly.
8) Either use tick marks or delete from the x-axis but keep it consistent.
Reply: It was revised accordingly.
9) In Figure 3B, ANOVA should be used to analyze the data instead of t-test.
Reply: The surgery in sham group was different from others. Thus there were two factors in this experiment, surgery and treatment or dose. ANOVA could be done if nessesary for many subgroups (more than three). Considering that there were only three groups, the single factor for sham and model control group was surgery, hence t-test could be done for the data in these two groups. The single factor for model control and model+treatment group was treatment or dose, hence t-test could be done to analyze the difference.
10) Show double or triple co-labeling in an additional high magnification figure.
Reply: The original images were in high magnification. They will be provided for enquire as below.
Image here in attachment
(Our data from Figure 2 H-J)
11) Also, there is no t- or F-value mentioned in the text.
Reply: The related information was added accordingly.

Reviewer 4 Report
Reviewer #1: The manuscript “Involvement of DAAO overexpression in delayed hippocampal neuronal death” by Hao Liu, Jun-Tao Zhang, Chen-Ye Mou, Yue Hao, Wei Cui
This manuscript describes a relationship between an overexpression and activation of DAAO in astrocytes with decreased viability of neurons isolated with the hippocampus after transient ischemia result from cerebral artery occlusion (MCAO).
Major points
Introduction
It does not give sufficient background about function of the DAAO function in the brain. The statement “DAAO is involved” is so far insufficient. How can be involved in ischemic stroke? Or what is know up to now.
No convincing explanation why DAAO expression was assay in the hippocampus (the suggestion in the previous papers), since transient carotid artery closure ( MCAO) affected wider cortical and subcortical areas.
Methods
A lot of inaccuracies in description and English grammar flows
1. Isolation of the hippocampus from 1-day pops is very difficult. It is a lack of a description the hippocampus isolation.
2. Pentobarbital, not isoflurane, was used for anesthesia which stronger affects glutamate-mediated excitatory or GABA-mediated inhibitory inputs to CA1 neurons, and it could affect neuronal viability.
3. Do purified astrocytes were culture in the same flasks??? What was complete culture medium/DMEM ??? No information about manufacturers or catalog No.
Results
1. Hippocampal expression of DAAO should be first presented in whole brain slides to be sure a proper recognition of the hippocampus.
2. What chemical compound was measured in the DAAO activity assay which gives fluorescent emission at l396 nm after excitation l251 nm?
3. How was taken the tissue from the brain slices for WB at time-course assay??? Pooled from all animals in each group or it was taken individually from each slide?
4. In the figures description should be describe shortly the method. For example in method section WB for the time-course effect of MCAO. How was prepared samples for WB from transfected astrocytes??? What method was determined total proteins???
5. Figure 5: a statement about a dose-dependent D-Ala promotion of apoptosis is unauthorized, since only two concentration (0.3 and 1 mM) D-Ala was assayed. Which significance of the result presented in the graph A, B or M was analyzed by ANOVA which by Student`s t test??
6. Figure 6: If ANOVA with post-hoc test indicates p<0.05 for three tested compounds (SUN, SKF96365 and PBN) on neuron viability and LDH content in neuronal medium, why only TRP-calcium channel is suspected to participate in neuronal death?? Moreover, the statement that these channels are involved specifically in apoptosis activation only after neuron viability determination not activation of apoptosis (Annexin V immunofluorescence, caspase activation ect.) is unauthorized.
7. Primary neurons for the TUNEL assay were platelet per 1 x 105 in 96-wells. How long before adding ACM these neurons were cultured in the plates? In what condition??
Discussion
The function of DAAO is a degradation of D-amino acids, in between D-Serine, an important gliotransmitter, an endogenous agonist of the NMDA receptors. Transient cerebral ischemia is associated with elevation of D-serine concentration resulted from astrocyte activation. Hyperactivation of DAAO through metabolizing of important amino acidic transmitters has to have much wider effects in the brain functionality, not only in the hippocampus. This should be discussed.
The conclusion “for management of DAAO would be a beneficial target after ischemic stroke” is too far gone. It's just one of the enzymes activated during reperfusion after stroke, and probably not the greatest source of H202.
Minor points
The manuscript is written very vague and generally.
For example in Figure 4: “the date expression methods is mean ± SEM (n=6)”. The figure subparts L depicts a number of normal neurons with Nissl bodes and M – the ration of injures with pyknotic nuclei to total cells. What does mean n=6??? A number of counting areas in the immunostaining slide or number of slides per group??
Author Response
Major points
Introduction
It does not give sufficient background about function of the DAAO function in the brain. The statement “DAAO is involved” is so far insufficient. How can be involved in ischemic stroke? Or what is know up to now.
Reply: Much more information of DAAO was added in introduction section.
No convincing explanation why DAAO expression was assay in the hippocampus (the suggestion in the previous papers), since transient carotid artery closure ( MCAO) affected wider cortical and subcortical areas.
Reply: It is well known that MCAO-induced ischemic stroke could lead to delayed death of hippocampal neurons (PMID: 15364023, PMID: 21212608), which is related to the dysfunction of astrocytes (PMID: 15364023, PMID: 32689584, PMID: 10320781). Besides, hippocampal pyramidal cells are the main projection neurons of cerebral cortex, thus MCAO mouse model is suitable to be studied on the underlying mechanism of delayed neuronal death via astrocyte-neuron interactions.
Methods
A lot of inaccuracies in description and English grammar flows
Reply: Many thanks for your comments. We will try to polish the text and make statement concise.
- Isolation of the hippocampus from 1-day pops is very difficult. It is a lack of a description the hippocampus isolation.
Reply: The process of hippocampus isolation was described accordingly in the 2.9 subsection.
- Pentobarbital, not isoflurane, was used for anesthesia which stronger affects glutamate-mediated excitatory or GABA-mediated inhibitory inputs to CA1 neurons, and it could affect neuronal viability.
Reply: Many thanks for your comments. The pentobarbital anesthesia indeed affects the neuronal viability in a certain degree. Fortunately, only a single injection of pentobarbital was used in moderate dose in this acute surgery. To reduce the potential impact of pentobarbital, the time course of many indicators were studied instead. And in vitro studies were also performed to avoid the influence of pentobarbital and others.
- Do purified astrocytes were culture in the same flasks??? What was complete culture medium/DMEM ??? No information about manufacturers or catalog No.
Reply: The related information was added accordingly.
Results
- Hippocampal expression of DAAO should be first presented in whole brain slides to be sure a proper recognition of the hippocampus.
Reply: It was shown in figure 4A.
- What chemical compound was measured in the DAAO activity assay which gives fluorescent emission at l396 nm after excitation l251 nm?
Reply: Kynurenic acid (KYNA) was measured as the product of substrate D-kynurenine after catalytic reaction of DAAO.
- How was taken the tissue from the brain slices for WB at time-course assay??? Pooled from all animals in each group or it was taken individually from each slide?
Reply: The mice on Day 5 would be performed with ischemic surgery 5 days before others, deduce the rest in this way, finally all mice would be sacrificed for WB assay. The hippocampal sample for each mouse will be extracted and prepared individually.
- In the figures description should be describe shortly the method. For example in method section WB for the time-course effect of MCAO. How was prepared samples for WB from transfected astrocytes??? What method was determined total proteins???
Reply: The related information was added in the legend of figure 5.
- Figure 5: a statement about a dose-dependent D-Ala promotion of apoptosis is unauthorized, since only two concentration (0.3 and 1 mM) D-Ala was assayed. Which significance of the result presented in the graph A, B or M was analyzed by ANOVA which by Student`s t test??
Reply: Thanks for your suggestion. The phrase “in a dose dependent manner” was deleted. It was confirmed in the legend of figure 5 that data in A, B were analyzed by Student’s t test, and data in M were analyzed by ANOVA.
- Figure 6: If ANOVA with post-hoc test indicates p<0.05 for three tested compounds (SUN, SKF96365 and PBN) on neuron viability and LDH content in neuronal medium, why only TRP-calcium channel is suspected to participate in neuronal death?? Moreover, the statement that these channels are involved specifically in apoptosis activation only after neuron viability determination not activation of apoptosis (Annexin V immunofluorescence, caspase activation ect.) is unauthorized.
Reply: The related information about TRP-calcium channel was added in the content of 3.5 subsection. The data of figure 6 was mainly focused on the underlying mechanism of DAAO-induced cytotoxicity. These studies of cellular apoptosis are also good suggestions.
- Primary neurons for the TUNEL assay were platelet per 1 x 105 in 96-wells. How long before adding ACM these neurons were cultured in the plates? In what condition??
Reply: The related information was added in the 2.12 subsection.
Discussion
The function of DAAO is a degradation of D-amino acids, in between D-Serine, an important gliotransmitter, an endogenous agonist of the NMDA receptors. Transient cerebral ischemia is associated with elevation of D-serine concentration resulted from astrocyte activation. Hyperactivation of DAAO through metabolizing of important amino acidic transmitters has to have much wider effects in the brain functionality, not only in the hippocampus. This should be discussed.
The conclusion “for management of DAAO would be a beneficial target after ischemic stroke” is too far gone. It's just one of the enzymes activated during reperfusion after stroke, and probably not the greatest source of H202.
Reply: Yes, we agree with your suggestion. The related content was revised in the section of conclusion.
Minor points
The manuscript is written very vague and generally.
Reply: Many thanks for your suggestion. We made a major revision totally.
For example in Figure 4: “the date expression methods is mean ± SEM (n=6)”. The figure subparts L depicts a number of normal neurons with Nissl bodes and M – the ration of injures with pyknotic nuclei to total cells. What does mean n=6??? A number of counting areas in the immunostaining slide or number of slides per group??
Reply: It indicated the number of slides in each group.

Round 2
Reviewer 1 Report
I have read the revised manuscript of Liu et al. submitted to Cells. The manuscript developed much, but is still not suitable for publication. Many major and some minor points are listed below. Furthermore, the language of the manuscript is still poor, and it includes a lot of grammar errors and bad phrasings. In summary, I suggest to thoroughly rewrite the manuscript by correcting the below mentioned issues regarding technical details and composition of the report. The manuscript should be resubmitted after such a substantial revision.
The control groups are still not clear. It seems to me that the 0 h group are mice that underwent MCAO surgery (ischemia/reperfusion). And there was also a sham group mentioned that was subjected to the same surgery without blocking the carotid artery. However, it is not clear why the Authors did not include the sham group in comparisons, e.g., on the graphs of Figure 2. The correct control group for these analyses would be the sham group.
Another confusing part of the explanation of experimental groups is the following:
„the relationship between DAAO expression and pathological changes in MCAO mice on Day 0, 1, 3 and 5 after ischemia/reperfusion” – this implies that the experimental groups are mice that were sacrificed on Day 0, 1, 3, 5. All of these are animals that underwent MCAO ischemia/reperfusion. (See also later in the Results: „quantified at four time points (0 hour, 24 hours, 3 days and 5 days) post MCAO surgery”.
Next:
„The normal mice were defined as MCAO mice on Day 0” – First: what does it mean that „normal mice”? It is not an appropriate or exact scientific definition. Second: this implies that the „Day 0” group are mice in that MCAO was not induced (i.e., sham operated mice) - in contrast with the above description.
Then, for the next experiment:
„mice were divided into sham group, MCAO control (MCAO mice were treated with saline only) and MCAO treatment group (MCAO mice were administrated with 10 mg/kg DAAO inhibitor SUN by intravenous multiple twice-daily injections)” – The groups were again differently defined than in the first experiment, and this substantially disturb the understanding of the reporting of the experiments.
These confusions in the naming of groups and in the comparisons (proper control groups) must be clarified and corrected.
„The mice in sham group suffered from the same surgery procedures but without inserting the monofilament into the common carotid artery to induce ischemia/reperfusion” – This sentence is redundant, the sham operation has already been defined earlier.
The explanation of the immunofluorescence stainings is not well ordered. The staining procedure is intermitted by this sentence: „Hippocampal DAAO was immuno-colocalized with microglia, astrocytes, or neurons and visualized under a TCS SP8 confocal microscope (Leica Microsystems, Wetzlar, Germany).” It is not clear whether the following sentences are the continuation of the procedure written before, or is it another staining protocol? It is also not clear from the explanation that different cytological markers were applied on different sections in co-localization with DAAO.
The formula describing the calculation of DAAO activity in the KYNA assay is hard to understand because of the long explanations in parentheses. Please, restructure.
TUNEL and Nissl-staining: this section would fit better after the immunofluorescence section.
„TUNEL immunostaining was studied according to the instruction of TUNEL cell apoptosis detection kit” – Please, specify the calculation method.
„LDH cytotoxicity was executed by LDH assay kit following the manufacturer's instructions” – a brief description of the method would be desirable.
„Statistical analysis” section lacks the specification of the applied statitical tests (e.g., ANOVA, t-test, post-hoc tests). Please, specify them for each experiments.
On Figure 1, it is hard to evaluate the extent of co-localization of DAAO with specific cellular markers. Larger photomicrographs and/or some kind of quantification (e.g., number of co-localized pixels) would help, as well as showing cellular markers’ and DAAO's signal also on separate pictures would clarify the presented results.
Again, photomicrographs on Figure 2 are also hard to be evaluated by the reader. This figure could be devided into two figures by showing the immunofluorescence results (pictures and graphs) on a separate figure. Thus, Authors could show nestin/GFAP and DAAO channels on separate pictures, and could increase the size of photos.
„As shown in Figure 2B-F, the intensity of nestin/DAAO strikingly increased by more than 11.0 times on Day 5 after MCAO onset (P=0.0001, by one-way ANOVA with post-hoc Student-Newman-Keuls test).” It is not clear from this (and previous) sentence whether the increase is in the area of nestin/DAAO double-labeling in the whole scanned area or the proportion of nestin/DAAO co-localization to the nestin-labelled area. If the former, than how was the intensity of pictures normalized/standardized between samples and how was the threshold between labelled/non-labelled pixels defined?
In Section 3.3. „compared to model group” is a strange phrasing for the MCAO+saline group. Use the specific term instead.
Results 3.3: The text states: „It indicated that DAAO was mainly expressed in GFAP-positive astrocytes in hippocampal CA2 (Figure 4A-D).” From the photomicrographs this is not evident for the reader. On Fig 4A, DAAO expression is also abundant e.g., in the dentate gyrus. Also, a mark on the photos that shows what is defined as CA2 by the Authors would help in the evaluation of the expression patterns.
After reporting the localization of DAAO/GFAP and TUNEL labellings, Authors suddenly switch to the reporting of the effects of SUN treatment: „The number of total neurons in model group treated with saline was deceased by 29.7% compared to sham group, which was alleviated by administration of SUN (10 mg/kg) with elevation of 17.3% (t=3.96, P=0.0027, by unpaired and two-tailed Student t-test; Figure 4L).” First, no photomicrographs of sham and MCAO+SUN samples have been shown on Figure 4 for the purpose of comparison. Second, it is not clear, what is the region where the cell numbers were counted (also CA2 or another region?). („model group” is again must be written more specifically)
„We did not set up the sham group in a part of subsections, which was a limitation of this study.” – I think, if an important control group is missing than it is not enough to point it out as a limitation, but it should be complemented.
Author Response
The control groups are still not clear. It seems to me that the 0 h group are mice that underwent MCAO surgery (ischemia/reperfusion). And there was also a sham group mentioned that was subjected to the same surgery without blocking the carotid artery. However, it is not clear why the Authors did not include the sham group in comparisons, e.g., on the graphs of Figure 2. The correct control group for these analyses would be the sham group.
Another confusing part of the explanation of experimental groups is the following:
„the relationship between DAAO expression and pathological changes in MCAO mice on Day 0, 1, 3 and 5 after ischemia/reperfusion” – this implies that the experimental groups are mice that were sacrificed on Day 0, 1, 3, 5. All of these are animals that underwent MCAO ischemia/reperfusion. (See also later in the Results: „quantified at four time points (0 hour, 24 hours, 3 days and 5 days) post MCAO surgery”.
Next:
„The normal mice were defined as MCAO mice on Day 0” – First: what does it mean that „normal mice”? It is not an appropriate or exact scientific definition. Second: this implies that the „Day 0” group are mice in that MCAO was not induced (i.e., sham operated mice) - in contrast with the above description.
Then, for the next experiment:
„mice were divided into sham group, MCAO control (MCAO mice were treated with saline only) and MCAO treatment group (MCAO mice were administrated with 10 mg/kg DAAO inhibitor SUN by intravenous multiple twice-daily injections)” – The groups were again differently defined than in the first experiment, and this substantially disturb the understanding of the reporting of the experiments.
These confusions in the naming of groups and in the comparisons (proper control groups) must be clarified and corrected.
Reply: Many thanks for your concern on the definition of MCAO mice on Day 0. In our opinion, the main concern of the experimental design is the time point, no matter MCAO or sham surgery, once the surgery was conducted, then the time point is not zero anymore. In these experiments concerning time points, the experimental design is identical. Besides, in the treatment studies, the sham and MCAO surgery were applied to compare the difference between ischemia/reperfusion and surgical itself. According to our former reports, the results of TTC staining showed that there was no infarct size in sham group (PMID: 31309542). In this study, the results of Nissl staining also demonstrated that there was few injured cells in sham group. For the difference of GFAP-positive activated astrocytes in sham and MCAO group, it was studied extensively in other studies that there was rare GFAP-positive astrocytes in sham as shown below.
(REFERENCE from figure 2 in PMID: 22940431)
„The mice in sham group suffered from the same surgery procedures but without inserting the monofilament into the common carotid artery to induce ischemia/reperfusion” – This sentence is redundant, the sham operation has already been defined earlier.
Reply: It was deleted.
The explanation of the immunofluorescence stainings is not well ordered. The staining procedure is intermitted by this sentence: „Hippocampal DAAO was immuno-colocalized with microglia, astrocytes, or neurons and visualized under a TCS SP8 confocal microscope (Leica Microsystems, Wetzlar, Germany).” It is not clear whether the following sentences are the continuation of the procedure written before, or is it another staining protocol? It is also not clear from the explanation that different cytological markers were applied on different sections in co-localization with DAAO.
Reply: Many thanks for your attention. It should be a general statement of the immunofluorescence stainings. This sentence was moved to the end of this paragraph.
The formula describing the calculation of DAAO activity in the KYNA assay is hard to understand because of the long explanations in parentheses. Please, restructure.
Reply: The describing of formula was restructured accordingly.
TUNEL and Nissl-staining: this section would fit better after the immunofluorescence section.
„TUNEL immunostaining was studied according to the instruction of TUNEL cell apoptosis detection kit” – Please, specify the calculation method.
Reply: The quantitative assay was only studied in Nissl-staining test. The related calculation method was added accordingly.
„LDH cytotoxicity was executed by LDH assay kit following the manufacturer's instructions” – a brief description of the method would be desirable.
Reply: It was added accordingly.
„Statistical analysis” section lacks the specification of the applied statitical tests (e.g., ANOVA, t-test, post-hoc tests). Please, specify them for each experiments.
Reply: The related information was added accordingly.
On Figure 1, it is hard to evaluate the extent of co-localization of DAAO with specific cellular markers. Larger photomicrographs and/or some kind of quantification (e.g., number of co-localized pixels) would help, as well as showing cellular markers’ and DAAO's signal also on separate pictures would clarify the presented results.
Reply: The related figures were revised accordingly.
Again, photomicrographs on Figure 2 are also hard to be evaluated by the reader. This figure could be devided into two figures by showing the immunofluorescence results (pictures and graphs) on a separate figure. Thus, Authors could show nestin/GFAP and DAAO channels on separate pictures, and could increase the size of photos.
Reply: The related figures were revised accordingly. And the size of photos in GFAP/DAAO channels was enlarged accordingly.
„As shown in Figure 2B-F, the intensity of nestin/DAAO strikingly increased by more than 11.0 times on Day 5 after MCAO onset (P=0.0001, by one-way ANOVA with post-hoc Student-Newman-Keuls test).” It is not clear from this (and previous) sentence whether the increase is in the area of nestin/DAAO double-labeling in the whole scanned area or the proportion of nestin/DAAO co-localization to the nestin-labelled area. If the former, than how was the intensity of pictures normalized/standardized between samples and how was the threshold between labelled/non-labelled pixels defined?
Reply: The quantified area fraction was represented by the ratio of colocalized area to total area of the whole scanned area. The intensity of pictures in samples was standardized to the consistent level by adjusting brightness to the same value. The threshold between labelled/non-labelled pixels was defined by operator in a fix value as shown below.
In Section 3.3. „compared to model group” is a strange phrasing for the MCAO+saline group. Use the specific term instead.
Reply: It was revised.
Results 3.3: The text states: „It indicated that DAAO was mainly expressed in GFAP-positive astrocytes in hippocampal CA2 (Figure 4A-D).” From the photomicrographs this is not evident for the reader. On Fig 4A, DAAO expression is also abundant e.g., in the dentate gyrus. Also, a mark on the photos that shows what is defined as CA2 by the Authors would help in the evaluation of the expression patterns.
Reply: It was revised accordingly.
After reporting the localization of DAAO/GFAP and TUNEL labellings, Authors suddenly switch to the reporting of the effects of SUN treatment: „The number of total neurons in model group treated with saline was deceased by 29.7% compared to sham group, which was alleviated by administration of SUN (10 mg/kg) with elevation of 17.3% (t=3.96, P=0.0027, by unpaired and two-tailed Student t-test; Figure 4L).” First, no photomicrographs of sham and MCAO+SUN samples have been shown on Figure 4 for the purpose of comparison. Second, it is not clear, what is the region where the cell numbers were counted (also CA2 or another region?). („model group” is again must be written more specifically)
Reply: The quantitative assay was only studied in Nissl-staining test. The photomicrographs of Figure 4 I-K were focused on CA2 region, which was informed in legend.
„We did not set up the sham group in a part of subsections, which was a limitation of this study.” – I think, if an important control group is missing than it is not enough to point it out as a limitation, but it should be complemented.
Reply: Many thanks for your concern. As we explained in the first comment, extensive studies studied the difference of GFAP-positive activated astrocytes in sham and MCAO group. Besides, both sham group and model control group were included in all treatment studies.

Reviewer 2 Report
Review of revised version of Liu et al: Involvement of DAAO overexpression in delayed hippocampanl neuronal death
The revised manuscript has addressed some of the points raised regarding the original version of the manuscript, but several issues still remain. In particular, the manuscript still needs to be assessed for grammatical errors – ideally by a native speaker. In addition, although the introduction has been expanded, it still lacks some important background, a clear outline of what the authors have already published on this topic, and what the specific aims of this study are. The confusion regarding the sham and the 0 hour time-points remains unaddressed.
Specific comments:
Abstract. It states ‘…with peak disease increase on day 5 after surgery, followed by successive neurobehavioural deficits’. Since no time-points were assessed later than day 5, it cannot be stated that this is the peak. Similarly, the neurobehavioural deficits were also not assessed after this timepoint, and so cannot be said to follow successively.
Introduction. What are the physiological roles of DAAO? What were the explicit aims of this study? In response to the request for the authors to clarify the aim of this study, they included the phrase ‘…in the aspect of astrocyte-neuron interactions’ which has not made anything clearer to me. A clearer explanation was made in the author’s response letter that although they have previously shown DAAO to be upregulated following MCAO and to be expressed predominantly in astrocytes, in this study they have focussed on the hippocampus. This should be added to the introduction section (along with reasoning for looking at the hippocampus). Similarly, an explicit statement that they previously showed that SUN inhibition of DAAO reduced the neurological deficit following MCAO, but the mechanism remains unclear is needed. It would then follow that the aim of this study was subsequently to investigate whether astrocytic DAAO in the hippocampus influences the neurological deficit through its effects on neuronal survival.
Methods. This section is vastly improved by the inclusion of more details, however, some of the corrections that have been made are worse than the original phrases. Some examples include:
- “All the drugs and reagents could be dissolved in buffer solution” was better as “All the drugs and reagents were dissolved in saline”. Otherwise, which buffer solutions were used?
- “The mice were feeded with enough food and water” was better as “The mice were reared in cages with free access to food and water”
- “…a silicone tip was infixed into common cerebral artery” … was better as “…a silicone tip was inserted into (the) common cerebral artery”
- “The mice in sham group were suffered from the …” was better as “The mice in (the) sham group were subjected to the …”, etc
-
Results
Figure 2. The confusion regarding sham and 0 hours has not been addressed. The figure continues to state 0 hours, but the legend to Figure 2 says that the statistics were made in comparison to a sham group. If these are referring to the same thing (i.e. the sham had pseudo-surgery but then taken immediately into experiments), then please adopt the same single description (either sham or 0 hours). Otherwise, it reads as though data is discussed which has not been presented and should be included.
The same issue goes for Figures 3 and 4 where a sham group is mentioned. Is this also time-point 0 hours? In which case I would hesitate to consider it a proper sham since the effect of the pseudo-surgery has not been assessed at comparable time-points, and therefore the changes being shown cannot be attributed specifically to the occlusion and not the associated surgery. If the sham was kept for 5 days post pseudo-surgery, then please clarify in the text.
Figure 3. the authors explain in their response that whereas they published previously using the Clark score, here they have used the Zea-Longa neurological grading score. This should be mentioned in the results text, that this experiment confirmed previously established data that treatment with SUN reduced the neurological deficit following MCAO, but was assessed using a different score. It should be transparent to the reader when data is confirming established concepts, or whether the findings are novel.
Discussion. The authors have reworded the sentence ‘It was identical with our work in study …’ please change to ‘It was identical with our work presented in THIS study ..” to clarify which work is being referred to. The following sentence only states that it was previously shown by the group that ischaemia induced astrocytic scarring – they should mention that this previous study also showed DAAO to be expressed specifically in the reactive astrocytes, too.
Author Response
The revised manuscript has addressed some of the points raised regarding the original version of the manuscript, but several issues still remain. In particular, the manuscript still needs to be assessed for grammatical errors – ideally by a native speaker. In addition, although the introduction has been expanded, it still lacks some important background, a clear outline of what the authors have already published on this topic, and what the specific aims of this study are. The confusion regarding the sham and the 0 hour time-points remains unaddressed.
Specific comments:
Abstract. It states ‘…with peak disease increase on day 5 after surgery, followed by successive neurobehavioural deficits’. Since no time-points were assessed later than day 5, it cannot be stated that this is the peak. Similarly, the neurobehavioural deficits were also not assessed after this timepoint, and so cannot be said to follow successively.
Reply: It was revised accordingly.
Introduction. What are the physiological roles of DAAO? What were the explicit aims of this study? In response to the request for the authors to clarify the aim of this study, they included the phrase ‘…in the aspect of astrocyte-neuron interactions’ which has not made anything clearer to me. A clearer explanation was made in the author’s response letter that although they have previously shown DAAO to be upregulated following MCAO and to be expressed predominantly in astrocytes, in this study they have focussed on the hippocampus. This should be added to the introduction section (along with reasoning for looking at the hippocampus). Similarly, an explicit statement that they previously showed that SUN inhibition of DAAO reduced the neurological deficit following MCAO, but the mechanism remains unclear is needed. It would then follow that the aim of this study was subsequently to investigate whether astrocytic DAAO in the hippocampus influences the neurological deficit through its effects on neuronal survival.
Reply: Many thanks for your suggestion. The physiological roles of DAAO were introduced in introduction section. The related information was also added.
Methods. This section is vastly improved by the inclusion of more details, however, some of the corrections that have been made are worse than the original phrases. Some examples include:
- “All the drugs and reagents could be dissolved in buffer solution” was better as “All the drugs and reagents were dissolved in saline”. Otherwise, which buffer solutions were used?
- “The mice were feeded with enough food and water” was better as “The mice were reared in cages with free access to food and water”
- “…a silicone tip was infixed into common cerebral artery” … was better as “…a silicone tip was inserted into (the) common cerebral artery”
- “The mice in sham group were suffered from the …” was better as “The mice in (the) sham group were subjected to the …”, etc
Reply: It was revised accordingly.
Results
Figure 2. The confusion regarding sham and 0 hours has not been addressed. The figure continues to state 0 hours, but the legend to Figure 2 says that the statistics were made in comparison to a sham group. If these are referring to the same thing (i.e. the sham had pseudo-surgery but then taken immediately into experiments), then please adopt the same single description (either sham or 0 hours). Otherwise, it reads as though data is discussed which has not been presented and should be included.
Reply: It was revised accordingly.
The same issue goes for Figures 3 and 4 where a sham group is mentioned. Is this also time-point 0 hours? In which case I would hesitate to consider it a proper sham since the effect of the pseudo-surgery has not been assessed at comparable time-points, and therefore the changes being shown cannot be attributed specifically to the occlusion and not the associated surgery. If the sham was kept for 5 days post pseudo-surgery, then please clarify in the text.
Reply: Many thanks for your comments. The explanation of sham group was added in the subsection of “2.3 The mouse MCAO model”.
Figure 3. the authors explain in their response that whereas they published previously using the Clark score, here they have used the Zea-Longa neurological grading score. This should be mentioned in the results text, that this experiment confirmed previously established data that treatment with SUN reduced the neurological deficit following MCAO, but was assessed using a different score. It should be transparent to the reader when data is confirming established concepts, or whether the findings are novel.
Reply: It was added accordingly.
Discussion. The authors have reworded the sentence ‘It was identical with our work in study …’ please change to ‘It was identical with our work presented in THIS study ..” to clarify which work is being referred to. The following sentence only states that it was previously shown by the group that ischaemia induced astrocytic scarring – they should mention that this previous study also showed DAAO to be expressed specifically in the reactive astrocytes, too.
Reply: It was added accordingly.
